# Dual orexin and MCH neuron-ablated mice display severe sleep attacks and cataplexy

**Chi Jung Hung**[1,2,3†], **Daisuke Ono**[1,2,3†], **Thomas S Kilduff**[4], **Akihiro Yamanaka**[1,2,3]*

[1]Department of Neuroscience II, Research Institute of Environmental Medicine, Nagoya University, Nagoya, Japan; [2]Department of Neural Regulation, Nagoya University Graduate School of Medicine, Nagoya, Japan; [3]CREST, JST, Honcho Kawaguchi, Saitama, Japan; [4]Center for Neuroscience, Biosciences Division, SRI International, Menlo Park, United States

**Abstract** Orexin/hypocretin-producing and melanin-concentrating hormone-producing (MCH) neurons are co-extensive in the hypothalamus and project throughout the brain to regulate sleep/ wakefulness. Ablation of orexin neurons decreases wakefulness and results in a narcolepsy-like phenotype, whereas ablation of MCH neurons increases wakefulness. Since it is unclear how orexin and MCH neurons interact to regulate sleep/wakefulness, we generated transgenic mice in which both orexin and MCH neurons could be ablated. Double-ablated mice exhibited increased wakefulness and decreased both rapid eye movement (REM) and non-REM (NREM) sleep. Double-ablated mice showed severe cataplexy compared with orexin neuron-ablated mice, suggesting that MCH neurons normally suppress cataplexy. Double-ablated mice also showed frequent sleep attacks with elevated spectral power in the delta and theta range, a unique state that we call 'delta-theta sleep'. Together, these results indicate a functional interaction between orexin and MCH neurons in vivo that suggests the synergistic involvement of these neuronal populations in the sleep/wakefulness cycle.

**\*For correspondence:**
yamank@riem.nagoya-u.ac.jp

†These authors contributed equally to this work

**Competing interests:** The authors declare that no competing interests exist.

## Introduction

The lateral hypothalamus (LH) has long been known to be involved in the regulation of sleep/wakefulness, feeding behavior and metabolism (*Théodoridès and Vetter, 1972*). Orexin/hypocretin- (encoded by the *Hcrt* gene) and melanin-concentrating hormone (MCH, encoded by the *Pmch* gene)-producing neurons are distributed within the LH; MCH neurons also extend caudally into the zona incerta. Orexin and MCH neurons project throughout the brain (*Bittencourt et al., 1992*; *Peyron et al., 1998*; *Nambu et al., 1999*) and are implicated in feeding and sleep/wakefulness (*Sakurai et al., 1998*; *Shimada et al., 1998*; *Chemelli et al., 1999*; *Verret et al., 2003*).

*Hcrt* (*Chemelli et al., 1999*) or *Hcrtr2* (*Willie et al., 2003*) gene knockout mice and orexin neuron-ablated mice (*Hara et al., 2001*; *Tabuchi et al., 2014*) display a narcolepsy-like phenotype. Narcolepsy is a chronic sleep disorder (*Mahoney et al., 2019*) caused by the specific degeneration of orexin neurons by the immune system (*Peyron et al., 2000*; *Latorre et al., 2018*). Narcolepsy patients have characteristic symptoms including excessive daytime sleepiness, hallucinations and cataplexy, a sudden loss of muscle tone triggered by positive emotions such as laughter (*American Sleep Disorders Association, 1990*; *Burgess and Scammell, 2012*). Optogenetic activation of orexin neurons induces wakefulness from sleep while optogenetic inhibition induces sleep from wakefulness (*Adamantidis et al., 2007*; *Tsunematsu et al., 2011*; *Schöne et al., 2012*; *Williams et al., 2019*). Together, these studies indicate that orexin neurons play an important role in the maintenance of wakefulness and prevent cataplexy induced by positive emotions.

The neuropeptide MCH was originally isolated from fish pituitary as a substance that controls skin pigmentation (*Kawauchi et al., 1983*). In mammals, MCH neurons are mainly distributed in the tuberal hypothalamus within which the orexin neurons are also located. Optogenetic activation of MCH neurons increases the total time in rapid eye movement (REM) sleep and reduces non-REM (NREM) sleep in mice (*Jego and Adamantidis, 2013*; *Konadhode et al., 2013*; *Tsunematsu et al., 2014*). Ablation of MCH neurons promotes wakefulness and decreases time in NREM sleep but has no effect on REM sleep (*Tsunematsu et al., 2014*). These observations suggest that MCH neurons are likely involved in the regulation of both NREM and REM sleep. We recently reported that REM sleep-active MCH neurons are involved in memory erasure during REM sleep (*Izawa et al., 2019*), further supporting the concept that MCH neurons are involved in multiple physiological functions (*Diniz and Bittencourt, 2017*; *Arrigoni et al., 2019*).

While the orexin and MCH neurons have different roles in the regulation of sleep/wakefulness (*Konadhode et al., 2014*), they have similar projection areas and receptor distributions (*Trivedi et al., 1998*; *Hervieu et al., 2000*; *Kilduff and de Lecea, 2001*; *Marcus et al., 2001*; *Saito et al., 2001*). Orexin and MCH neurons have also been reported to interact with each other. For example, application of orexin peptide increases spike frequency in MCH neurons in vitro (*van den Pol et al., 2004*) and optogenetic activation of orexin neurons inhibits MCH firing through GABA$_A$ receptors (*Apergis-Schoute et al., 2015*), while MCH reverses hypocretin-1-induced enhancement of action potentials in orexin neurons (*Rao et al., 2008*). Nevertheless, it is still unclear how interactions between orexin and MCH neurons contribute to sleep/wake regulation.

To understand the functional communication between orexin and MCH neurons and sleep/wakefulness regulation, we generated transgenic mice in which both orexin and MCH neurons were simultaneously ablated by tetracycline trans-activator (tTA)-induced expression of the diphtheria toxin A fragment (DTA). Analysis of the sleep patterns of orexin and MCH neuron double-ablated mice (OXMC mice) revealed that these mice exhibited very high levels of cataplexy and sleep/wake abnormalities. OXMC mice had increased wakefulness and profound reductions in REM sleep (particularly during the dark phase) and a corresponding increase in cataplexy, suggesting that MCH neurons have a suppressive role on cataplexy. OXMC mice also frequently showed short episodes of behavioral arrest with high δ and θ power. We defined this unique state as 'delta-theta sleep' (DT sleep) since this state was distinct from other states of sleep/wakefulness or cataplexy. Behavioral and pharmacological assessments showed some similarities as well as characteristic differences between DT sleep and cataplexy.

## Results

### Dual ablation of orexin and MCH neurons using the tetracycline transactivator (tTA)/TetO system

We previously reported that *Hcrt-tTA*; *TetO DTA* (OX) mice (*Tabuchi et al., 2014*) or *Pmch-tTA*; *TetO DTA* mice (*Tsunematsu et al., 2014*) enabled us to induce specific ablation of orexin neurons or MCH neurons in a timed-controlled manner, respectively. Here, we generated triple transgenic *Hcrt-tTA* (Tg/-); *Pmch-tTA* (Tg/-); *TetO DTA* (Tg/-) (OXMC) mice to enable simultaneous ablation of both orexin and MCH neurons (*Figure 1A*). DTA induces cell death by inactivating eukaryotic elongation factor 2 through ADP ribosylation, thereby blocking protein synthesis. DTA expression can be controlled by the presence or absence of doxycycline (DOX)-containing chow since the Tet-off system is used to control the timing of DTA expression (*Figure 1B*). In situ hybridization confirmed that MCH neurons and orexin neurons co-expressed *Slc17a6* (encoding vesicular glutamate transporter: vGlut2) mRNA but not *Slc32a1* (encoding vesicular inhibitory amino acid transporter: vGAT) mRNA. *Figure 1—figure supplement 1* indicates that most MCH and orexin neurons are co-localized with vGlut2 (MCH neurons: 96.3 ± 3.1%, n = 4; orexin neurons: 99.6 ± 0.2%, n = 5) but not with vGAT (MCH neurons: 5.1 ± 0.9%, n = 4; orexin neurons: 1.7 ± 0.9%, n = 3), suggesting that most MCH and orexin neurons are glutamatergic rather than GABAergic neurons as also suggested by others (*Mickelsen et al., 2017*; *Mickelsen et al., 2019*). Although MCH neurons and orexin neurons were a relatively minor subset of the LH glutamatergic neuron population (MCH neurons: 14.6 ± 2.4% (n = 4); orexin neurons: 23.6 ± 3.2% (n = 5)).

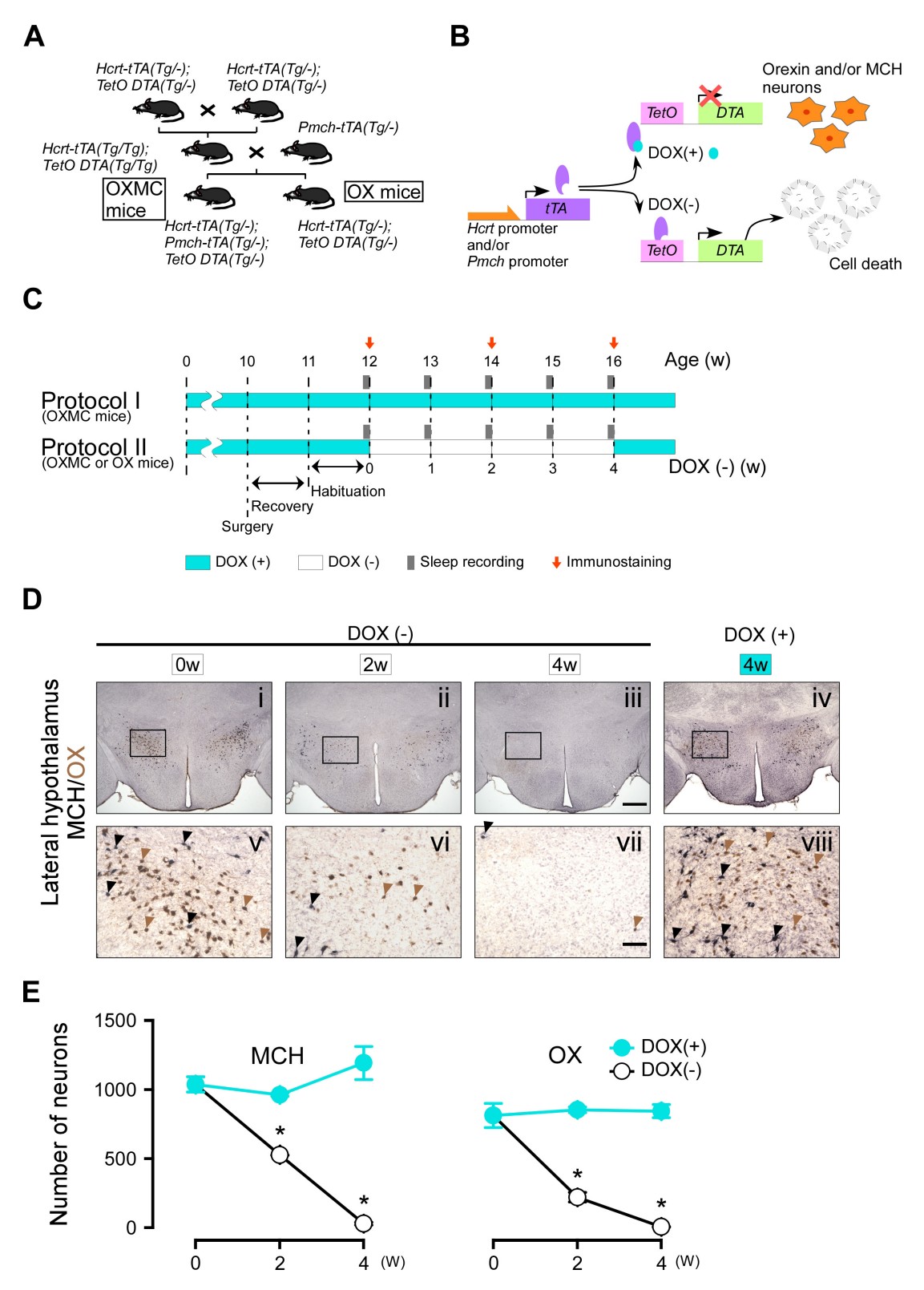

**Figure 1.** Dual ablation of orexin and MCH neurons using the tetracycline tTA/TetO system. (**A**) Schematic illustrating generation of *Hcrt-tTA* (Tg/-); *Pmch-tTA* (Tg/-); *TetO DTA* (Tg/-) mice. (**B**) Schematic showing use of the tetracycline-controlled Tet-off gene expression system. In the presence of DOX (light blue circle), DOX binds to tTA (purple oval) and DTA expression is blocked. The absence of DOX allows tTA to bind TetO, allowing transcription of the diphtheria toxin A (DTA) subunit which results in cell death. (**C**) Experimental protocols for sleep recording and immunostaining.
*Figure 1 continued on next page*

*Figure 1 continued*

Protocol I is the control (DOX(+)) condition; Protocol II is the experimental (DOX(-) for 4 weeks) condition. EEG and EMG surgeries were performed at age 10 weeks in the sleep-recording group; the mice used for immunostaining did not undergo surgery. Light blue and white bars represent the periods of DOX chow (DOX(+)) and normal chow (DOX(-)) availability, respectively. Gray bars indicate sleep recordings; red arrows indicate when mice were sacrificed for immunostaining. (D) Immunostaining of orexin (brown) and MCH (black) neurons in the LH at DOX(-) 0 week (i and v), 2 weeks (ii and vi), 4 weeks (iii and vii), and DOX(+) 4 weeks (iv and viii). Black and brown arrowheads indicate typical examples of orexin and MCH neurons, respectively. Panels v-viii are magnifications of the areas delineated by the squares in Panels i-iv. Scale bars: i-iv, 500 µm; v-viii, 100 µm. (E) The number of orexin and MCH neurons in OXMC mice from the DOX(+) and (-) conditions (n = 3–6). Values are mean ± SEM. *p<0.05 vs. DOX(+). Data were analyzed by unpaired *t* test.

The online version of this article includes the following source data and figure supplement(s) for figure 1:

**Source data 1.** Source data for *Figure 1E*.

**Figure supplement 1.** In situ RNA hybridization of orexin, MCH, vGlut2 and vGAT: orexin- and MCH-positive cells were colocalized with vGlut2 but not with vGAT.

**Figure supplement 1—source data 1.** Source data for *Figure 1—figure supplement 1–*.

*Figure 1C* illustrates the timing of DOX on (+), DOX off (-), sleep recording and immunostaining. In OXMC mice, both orexin and MCH neurons were simultaneously ablated by expressing DTA in the absence of DOX (*Figure 1B*). *Figure 1D* shows representative immunostaining of orexin and MCH neurons. The number of orexin neurons in OXMC mice in the DOX(+) condition at 0, 2 and 4 weeks was 810 ± 86 (100%, n = 3), 849 ± 20 (104.9%, n = 3) and 841 ± 47 (103.8%, n = 4), respectively. The number of MCH neurons at 0, 2 and 4 weeks was 1031 ± 54 (100%, n = 3), 958 ± 12 (92.9%, n = 3) and 1185 ± 118 (114.9%, n = 4), respectively. Conversely, the number of orexin neurons in OXMC mice in the DOX(-) condition at 2 and 4 weeks was 226 ± 34 (26.6%, n = 3, p<0.05 vs. DOX(+) 2 w) and 16 ± 3 cells (1.8%, n = 6, p<0.05 vs. DOX(+) 4 w), respectively. The number of MCH neurons at 2 and 4 weeks was 527 ± 1 (55.0%, n = 3, p<0.05 vs. DOX(+) 2 w) and 38 ± 10 cells (3.2%, n = 6, p<0.05 vs. DOX(+) 4 w), respectively (*Figure 1E*). These results confirmed that orexin and MCH cell bodies were ablated by removal of DOX from chow.

Orexin neurons and MCH neurons project widely throughout the brain. Dense projections of orexin neurons have been described in the locus coeruleus (LC) and raphe nucleus (*Peyron et al., 1998*; *Date et al., 1999*). MCH neurons densely innervate the medial septum (MS) and hippocampus (*Jego and Adamantidis, 2013*; *Izawa et al., 2019*). In OXMC mice maintained for 4 weeks in the DOX(-) condition, orexin and MCH nerve terminals in these projection sites were completely eliminated (*Figure 2*). These results confirmed that, not only the cell bodies, but also the projections of orexin and MCH neurons were eliminated in OXMC mice after DOX(-) for 4 weeks.

## Dual-ablated mice frequently displayed short behavioral arrests with high spectral power in the δ and θ bands of the EEG during wakefulness

Sleep and wakefulness was assessed in three groups of mice, OXMC DOX(-), OXMC DOX(+) and OX DOX(-) mice by EEG and EMG recording (*Figure 3A*). Spectral analyses of the EEG and EMG during wakefulness, NREM and REM sleep were indistinguishable across the three groups in the DOX(+) condition (before DOX removal, *Figure 4A and B*). After DOX removal, there was no difference in EEG spectrum and EMG integral of OXMC DOX(-) mice during any vigilance state as neuronal ablation proceeded (*Figure 4C, D and E*). These results indicate that neither orexin nor MCH neuron ablation affected the EEG power spectrum in any vigilance state.

Ablation of orexin neurons is known to induce narcolepsy-like symptoms, such as fragmentation of sleep and wakefulness, sleep onset REM sleep, and cataplexy-like behavioral arrests (*Hara et al., 2001*; *Tabuchi et al., 2014*). Since orexin neurons were ablated in both OX DOX(-) and OXMC DOX(-) mice, these mice showed cataplexy-like behavioral arrests but OXMC DOX(+) mice did not (*Figure 3B* and *Table 1*). In addition, OXMC DOX(-) mice frequently showed behavioral arrest episodes that were similar to cataplexy (*Video 1*) but which occurred after a sustained period of wakefulness. However, these behavioral arrest episodes were different from cataplexy based on a number of criteria (*Table 2*), particularly the EEG, which showed high amplitude spectral power in the δ and θ bandwidths during the behavioral arrests (*Figure 3C*). These episodes were defined as a sudden cessation of motor activity characterized by decreased EMG and relatively high-power ratios of δ and θ in the EEG, preceded by at least 40 s of wakefulness (10 epochs) and followed by a return

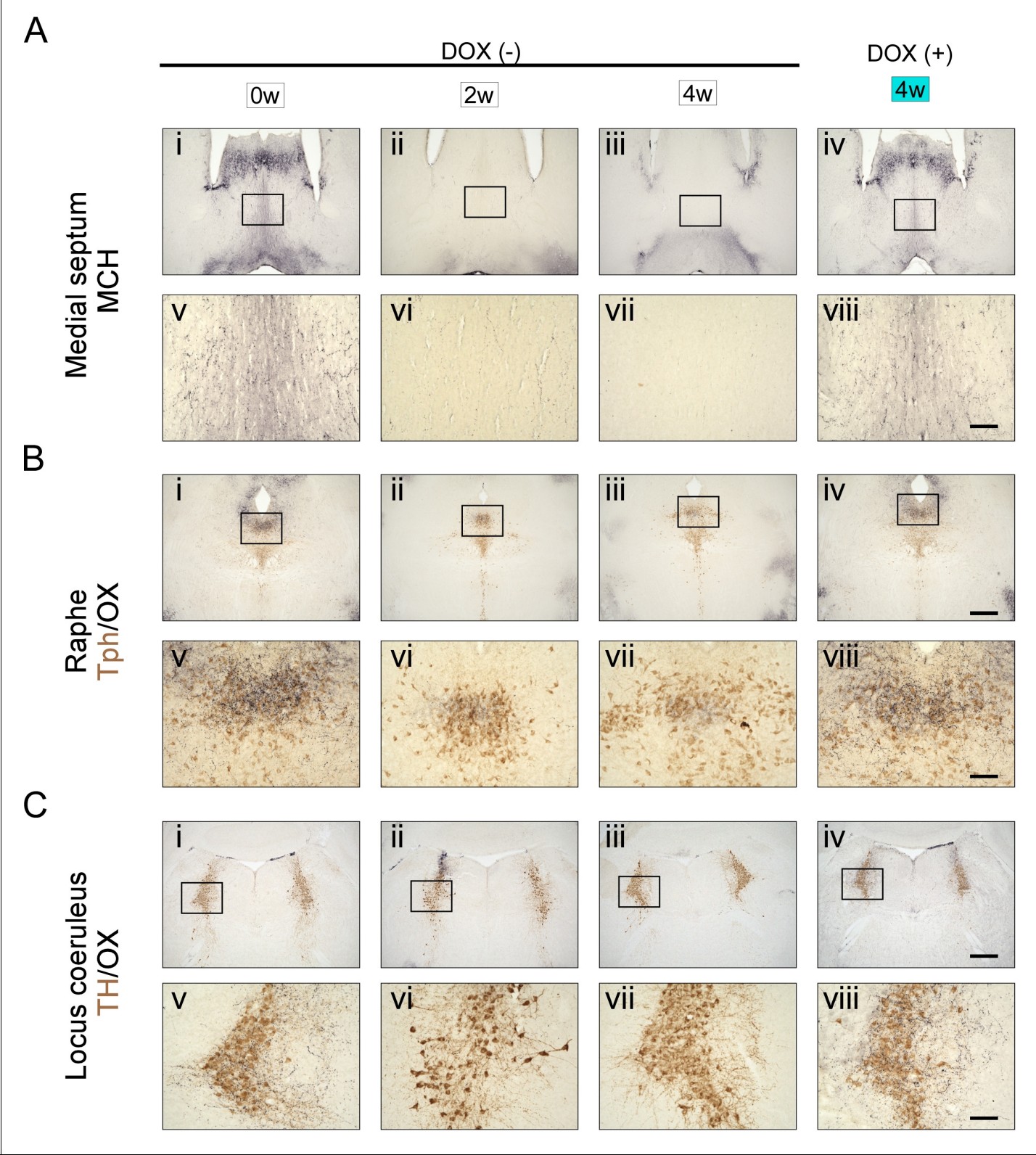

**Figure 2.** Time course of orexin and MCH nerve terminal ablation in projection areas. (**A**) Immunostaining of MCH nerve terminals (black) in the medium septum (MS). (**B**) Immunostaining of orexin nerve terminals (black) and tryptophan hydroxylase (Tph; brown) in the raphe nucleus. (**C**) Immunostaining of orexin nerve terminals (black) and tyrosine hydroxylase (TH; brown) in the locus coeruleus (LC). Photos indicate DOX(-) condition at

*Figure 2 continued on next page*

*Figure 2 continued*

week 0 (i and v), 2 weeks (ii and vi), 4 weeks (iii and vii) and the DOX(+) condition at 4 weeks (iv and viii). Panels v-viii are magnifications of the areas delineated by the squares in Panels **i-iv**. Scale bars: **i-iv**, 500 μm; v-viii, 100 μm.

to wakefulness. The relative power ratio of δ and θ bands during these arrests was significantly different from NREM sleep, REM sleep or cataplexy (*Figure 3D and E*). Consequently, we called the state during these arrests 'delta/theta sleep' (DT sleep). These results suggest that DT sleep is a novel brain state different from other states, as defined by EEG spectral characteristics.

## Effect of dual ablation of orexin and MCH neurons on sleep and wakefulness

The sleep/wakefulness patterns of OXMC mice were analyzed during the simultaneous ablation of both the orexin and MCH neurons. *Figure 5A* presents typical 24 hr hypnograms observed during the first 4 weeks of orexin and MCH neuron ablation. Sleep/wakefulness fragmentation was observed in both OXMC DOX(-) and OX DOX(-) mice by 1 week in the DOX(-) condition, particularly at the beginning of the dark phase (*Figure 5A*). Cataplexy was detected from 2 weeks DOX(-) onwards in both OXMC DOX(-) and OX DOX(-) mice (*Figure 5A, C and D*). DT sleep was only observed in OXMC mice; the first episode was detected at 1 week DOX(-), a week earlier than the initial observation of cataplexy at 2 weeks DOX(-). Since DT sleep often occurred during the early dark phase, *Figure 5B* presents an expanded hypnogram for 1 hr in the early dark phase at 4 weeks DOX(-) (the period enclosed by the magenta rectangle in *Figure 5A*). The mean DT sleep duration and number of DT bouts during the dark phase after 4 weeks of DOX(-) were 14.4 ± 0.7 s and 98.3 ± 6.2, respectively (*Figure 5C*). Although the total time and the number of DT sleep bouts observed during the light phase progressively increased by 4 weeks after DOX removal, neither the total time, mean bout duration nor the number of DT sleep bouts progressively increased during the dark phase as ablation proceeded (*Figure 5C and D*). Interestingly, the mean DT sleep bout duration appeared to plateau at ~15 s in both the light and dark phases by 2 weeks post-ablation (*Figure 5C and D*).

To clarify the functional role of MCH neurons, the total time, mean bout duration and the number of bouts of each vigilance state were compared at each stage of ablation. The total wakefulness time significantly increased in OXMC DOX(-) mice at 4 weeks after DOX removal in the both light and dark phases as compared with OX DOX(-) mice (*Figure 5C and D* and *Table 1*). Conversely, except for NREM sleep during the light period, OXMC DOX(-) mice exhibited a significant reduction in total REM and NREM sleep and the number of bouts in both light and dark phases at 4 weeks DOX(-) when compared with OX DOX(-) (*Figure 5C and D* and *Table 1*). The total time in cataplexy and mean cataplexy bout duration was greater in OXMC DOX(-) mice as compared with OX DOX(-) mice (*Figure 5C and D* and *Table 1*), indicating that the loss of MCH neurons exacerbated cataplexy symptomatology. *Table 1* summarizes the vigilance state and cataplexy characteristics in OXMC DOX(-) mice compared to OX DOX(-) mice. The differences in sleep/wakefulness phenotype between these two strains suggest that MCH neurons are part of a circuit that normally suppresses cataplexy.

## Behavioral assessment of DT sleep

To determine whether DT sleep is different from cataplexy or sleep, we performed tactile stimulation on OXMC DOX(-) mice during cataplexy, NREM or DT sleep. Although mice rarely responded to tactile stimulation during cataplexy (*Video 2*), they always responded to tactile stimulation during NREM or DT sleep by escaping the source of stimulation. *Figure 6A* illustrates that the probability of wakefulness was 33% after tactile stimulation during cataplexy compared to 100% in NREM and DT sleep. Mice could sense the touch of the brush and escaped during DT sleep and NREM sleep (*Video 3*). These results suggest that DT sleep differs from cataplexy and may be more similar to NREM sleep.

To further characterize DT sleep, we analyzed the position of mice in the home cage when DT sleep was initiated. The cage was divided into 4 areas, with a nest located in one of them (*Figure 6B*). NREM and REM sleep occurred with higher probability when mice were in the nest area

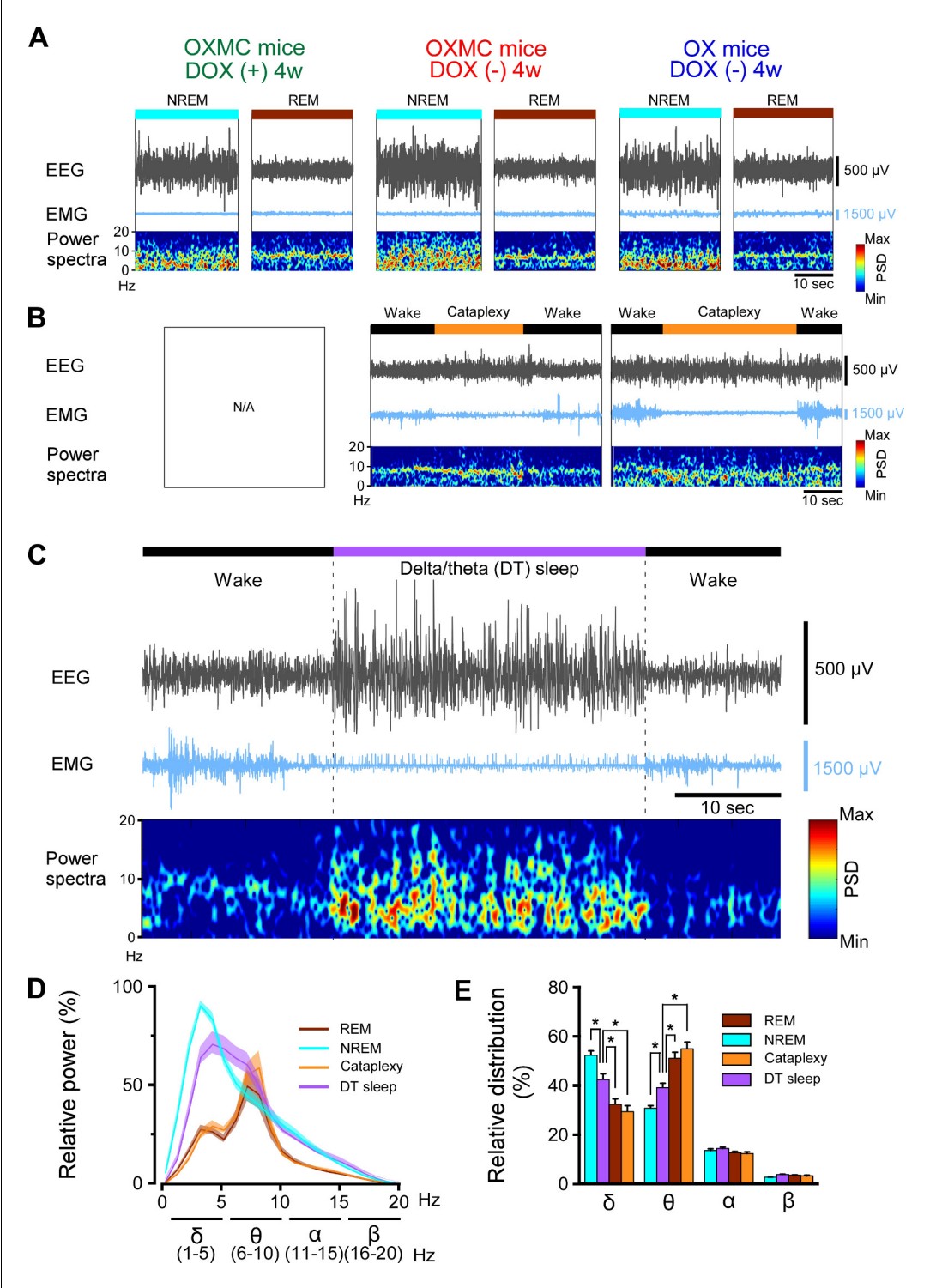

**Figure 3.** Criteria used to classify DT sleep. (**A**) Representative EEG and EMG traces and EEG power spectral density during NREM and REM sleep for OXMC mice after 4 weeks in the DOX(+) and DOX(-) conditions and for OX mice after 4 weeks in the DOX(-) condition. (**B**) Representative EEG and EMG traces and EEG power spectral density during cataplexy in OXMC DOX(-) and OX DOX(-) mice. (**C**) Representative EEG and EMG traces and EEG power spectral density during DT sleep in OXMC DOX(-) mice. DT sleep was characterized by low EMG amplitude with high EEG spectral power in the δ (1–5 Hz) and θ (6–10 Hz) bandwidths. (**D**) Relative EEG power during NREM, REM, cataplexy and DT sleep in OXMC DOX(-) mice (n = 6–8 from 0 to 4 weeks DOX(-). The greatest EEG power across all stages was set at 100% for each mouse. Values are presented as mean ± SEM. (**E**) Relative EEG power in the standard frequency bands (δ, θ, α and β) for each state. The sum of each band is 100%. Values are presented as mean ± SEM. *p<0.05. Data were

*Figure 3 continued on next page*

*Figure 3 continued*

analyzed by one-way ANOVA followed by the Bonferroni *post hoc* test. Despite comparable EMG levels, DT sleep has significantly greater spectral power in the δ range and less power in the θ range than either REM sleep or cataplexy.

The online version of this article includes the following source data for figure 3:

**Source data 1.** Source data for *Figure 3D and E*.

(NREM sleep: 76 ± 4.4%, REM sleep: 87 ± 7.0%, n = 6; *Figure 6B* bar graph). However, both DT sleep and cataplexy occurred in the nest area at almost a chance level (25%, *Figure 6B*). These results suggest that DT sleep occurs with a similar timing to cataplexy and, like cataplexy, may be a spontaneous, uncontrolled state.

We next categorized behaviors before cataplexy or DT sleep initiation by video analysis. The probability of running before cataplexy was significantly higher than running before DT sleep (*Figure 6C*). In comparison, the probability of grooming behavior before DT sleep was significantly higher than grooming before cataplexy (*Figure 6C*). Although the probabilities differed with some behaviors such as running and grooming, both DT sleep and cataplexy often occurred after running or digging. These results suggest that DT sleep can occur under situations similar to cataplexy.

Day- and night-time differences in cataplexy or DT sleep were also analyzed. As described previously (*Tabuchi et al., 2014*), orexin-ablated mice (OX DOX(-)) experienced more cataplexy during the dark phase than during the light phase (*Figure 6D*). In contrast, there was no substantial difference in cataplexy occurrence between the light and dark phases in OXMC DOX(-) mice, although DT sleep occurred more frequently in the dark phase compared to the light phase (*Figure 6D*). These results suggest that, even though MCH neuron ablation alone does not result in cataplexy, MCH neuron activity might be involved in the temporal regulation of cataplexy during both the light and dark phases.

## Pharmacological assessments of DT sleep

To further understand whether DT sleep is related to sleep or cataplexy, OXMC DOX(-) mice were exposed to chocolate to increase cataplexy, administered clomipramine to suppress cataplexy, or modafinil to promote wakefulness. After removing DOX chow for 4 weeks to generate dual orexin and MCH neuron ablation, DOX chow was replaced to stop further ablation and arrest further progress of sleep/wakefulness abnormalities (*Tabuchi et al., 2014*). The experimental protocols are illustrated in *Figures 7A*, *8A* and *9A*. Briefly, chocolate (1.7–1.9 g, milk chocolate, Meiji) was provided for 15 min just prior to dark onset (ZT12) for 3 days and sleep/wakefulness was analyzed on the third day. After a one-day interval, the vehicle, clomipramine or modafinil was administered followed by counterbalanced treatments on subsequent days. All administrations were performed just before dark onset.

Chocolate availability is known to increase the time in cataplexy and the number of cataplexy bouts in *Hcrt* knockout mice (*Burgess et al., 2013*; *Oishi et al., 2013*). Chocolate administration to OXMC DOX(-) mice significantly increased the total time in wakefulness and decreased REM and NREM sleep but did not affect total time or the number of bouts of DT sleep (*Figure 7B and C*). Clomipramine administration significantly decreased time in wakefulness and REM sleep and increased total time of NREM sleep. As expected, clomipramine also significantly inhibited cataplexy but did not affect DT sleep time, the number of bouts or their duration (*Figure 8B and C*). On the other hand, modafinil administration increased the total time of wakefulness, inhibited NREM and REM sleep and, as expected, did not affect cataplexy. Although the total time in DT sleep was not significantly affected by modafinil administration, the mean DT sleep bout duration was reduced (*Figure 9B and C*). These results suggest that the neural mechanisms underlying DT sleep might be distinct from those underlying cataplexy.

## Relationship between DT sleep and the transition from NREM to REM sleep

Since both δ and θ power in the EEG were high during DT sleep (*Figure 3D and E*), we evaluated the EEG during sleep/wake and wake/sleep transitions to compare EEG spectral characteristics. We found that the EEG spectrum in the transition from NREM to REM sleep was similar to DT sleep. We

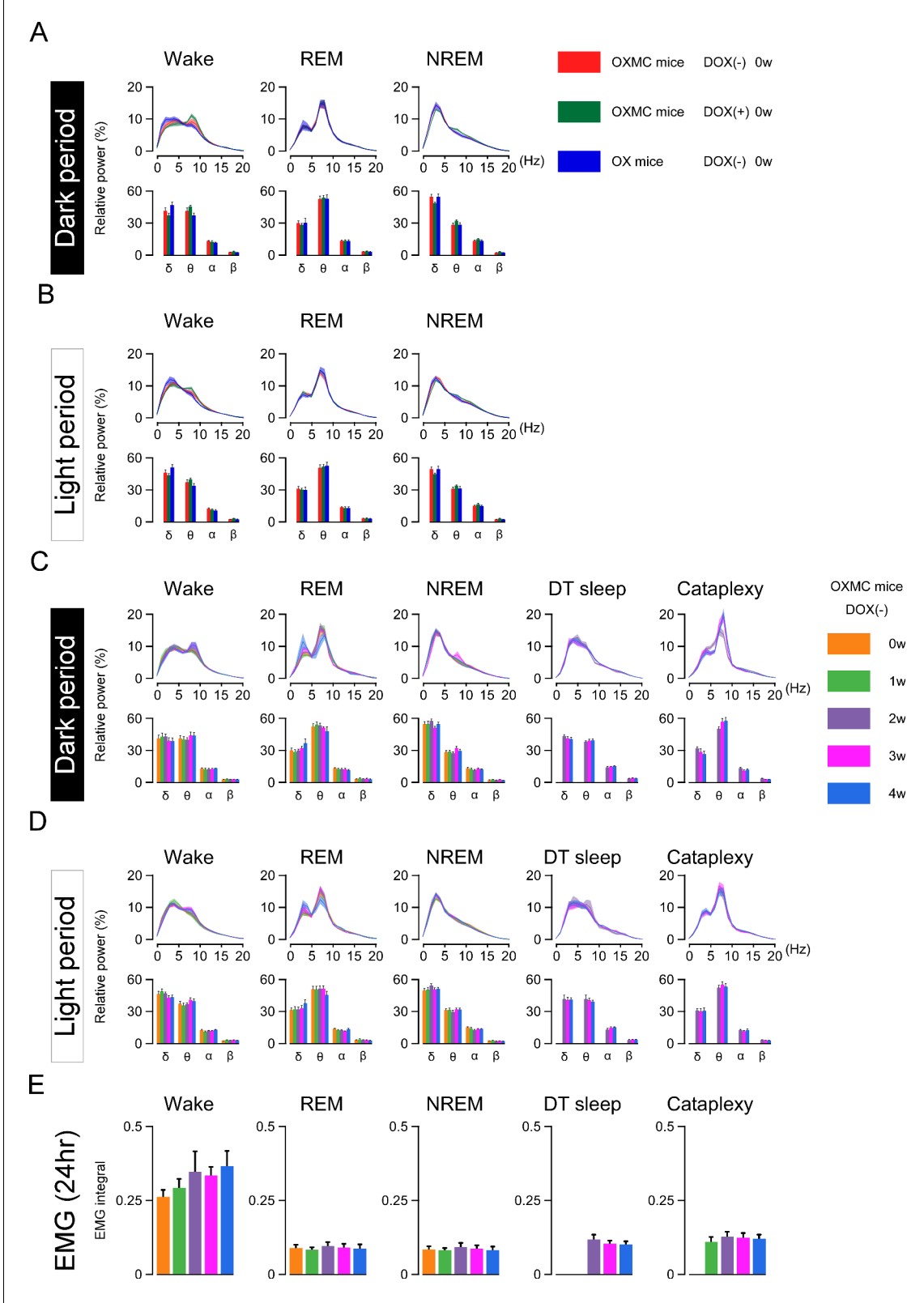

**Figure 4.** DOX removal from the diet in OXMC mice does not affect relative EEG power distribution or the EMG integral. (**A**) Relative EEG power in three strains of mice in the DOX(+) condition (before DOX removal) during the dark phase. Upper graphs show relative EEG power at 1 Hz resolution; lower graphs present the standard EEG frequency bands. (**B**) As in (**A**) but for the light phase. (**C**) Relative EEG power of OXMC mice measured during

*Figure 4 continued on next page*

*Figure 4 continued*

the dark phase from 0 to 4 weeks in the DOX(-) condition. Upper graphs show relative EEG power at 1 Hz resolution; lower graphs present the standard EEG frequency bands. (D) As in (C) but for the light phase. (E) EMG integral of OXMC mice in the DOX (-) condition. n = 6–8, Values are mean ± SEM.
The online version of this article includes the following source data for figure 4:

**Source data 1.** Source data for *Figure 4A-E*.

analyzed the last 3 epochs (12 s) in NREM sleep prior to transition to REM sleep in OXMC DOX(-) mice and compared them to the EEG power spectra in DT sleep (*Figure 10A*). The EEG spectra during the transition from NREM sleep to REM sleep was not altered by dual orexin and MCH neuron ablation (*Figure 10B*). Spectral power in the δ, θ, α and β bands of the EEG was indistinguishable between DT sleep and the NREM to REM sleep transition (*Figure 10C*). In addition, the 24 hr EMG integral did not differ between DT sleep and the NREM to REM transition (*Figure 10C*). These results indicated a similarity between DT sleep and the transition state from NREM to REM sleep.

## Discussion

To understand functional interactions between orexin and MCH neurons in the regulation of sleep and wakefulness, we generated dual orexin and MCH neuron-ablated mice and compared the resultant sleep abnormalities to those of orexin neuron-ablated mice. Double-ablated mice exhibited pronounced cataplexy and the total time in cataplexy and mean cataplexy bout duration were significantly increased, suggesting that MCH neurons normally have a suppressive role on cataplexy. Double-ablated mice also had exaggerated sleep abnormalities compared to singly-ablated or intact

**Table 1.** Comparison of parameters between orexin neuron-ablated mice and orexin MCH-blated mice.

| Parameter | Orexin neuron-ablated mice | Orexin and MCH-ablated mice | - fold Difference (OXMC/OX) | *P* value |
|---|---|---|---|---|
| Earliest detection timing of cataplexy | 2 weeks post-ablation | 2 weeks post-ablation | N/A | N/A |
| Time-of-day occurrence at 4 weeks | Both light and dark phases | Both light and dark phases | N/A | N/A |
| Total time in wakefulness during dark phase at 4 weeks | 418.9 ± 20.2 min | 571.0 ± 22.2 min | 1.36 | 1.4E-03 |
| Total time in NREM sleep during dark phase at 4 weeks | 258.3 ± 16.2 min | 90.5 ± 19.8 min | 0.35 | 2.8E-04 |
| Total time in REM sleep during dark phase at 4 weeks | 25.6 ± 0.9 min | 2.7 ± 1.7 min | 0.10 | 4.3E-06 |
| Total time in wakefulness during light phase at 4 weeks | 278.1 ± 19.7 min | 322.6 ± 7.3 min | 1.16 | 3.1E-02 |
| Total time in NREM sleep during light phase at 4 weeks | 389.8 ± 17.4 min | 340.1 ± 8.1 min | 0.87 | 1.6E-02 |
| Total time in REM sleep during light phase at 4 weeks | 46.3 ± 3.4 min | 20.8 ± 4.2 min | 0.45 | 2.7E-03 |
| Total time in cataplexy in dark phase at 4 weeks | 11.0 ± 0.6 min | 31.2 ± 3.6 min | 2.83 | 2.6E-03 |
| Number of cataplexy bouts in dark phase at 4 weeks | 14.8 ± 2.0 | 16.9 ± 2.1 | 1.14 | 0.53 |
| Mean bout duration in dark phase at 4 weeks | 53.3 ± 7.6 sec | 113.0 ± 9.6 sec | 2.12 | 2.2E-03 |
| Total time in cataplexy in light phase at 4 weeks | 5.4 ± 0.9 min | 24.2 ± 4.5 min | 4.45 | 1.4E-02 |
| Number of cataplexy bouts in light phase at 4 weeks | 8.3 ± 1.4 | 15.4 ± 3.1 | 1.87 | 0.13 |
| Mean bout duration in light phase at 4 weeks | 50.0 ± 11.4 sec | 96.3 ± 4.8 sec | 1.93 | 1.7E-03 |

The online version of this article includes the following source data for  Table 1:
Source data 1. Source data for *Table 1*.

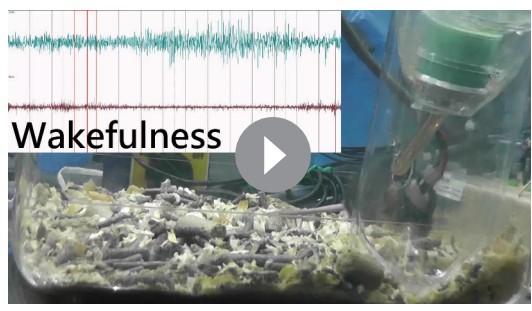

**Video 1.** Typical behavior for DT sleep in orexin- and MCH neuron-ablated mice.
https://elifesciences.org/articles/54275#video1

mice; specifically, increased time in wakefulness and decreased time in NREM and REM sleep. Double-ablated mice also exhibited a novel state that we called DT sleep, defined as an episode of sudden behavioral arrest of brief (~15 s) duration preceded by at least 40 s of wake and characterized by high δ and θ power in the EEG. Behavioral, electrophysiological and pharmacological assessments discriminated DT sleep from NREM, REM and cataplexy.

Cataplexy is well-known to be triggered by strong positive emotions (*Mignot, 1998*). Previously we showed that cataplexy is prevented by orexin but not by other co-localized neurotransmitters in the orexin neurons, such as glutamate and dynorphin (*Chowdhury et al., 2019*). Orexin neuron activity is thought to prevent cataplexy through activation of OX2R when positive emotion occurs since *Hcrt* or *Hcrtr2* gene knockout or ablation of orexin neurons induces cataplexy. The amygdala is implicated in the emotional stimulation that facilitates cataplexy (*Burgess et al., 2013*; *Hasegawa et al., 2014*; *Hasegawa et al., 2017*; *Mahoney et al., 2017*; *Snow et al., 2017*). Neurons in the amygdala project to the brainstem, which is crucial for the regulation of muscle tone and REM sleep (*Wallace et al., 1989*). Serotonergic neurons in the raphe nucleus projecting to the amygdala are thought to be the neural pathway responsible for suppression of cataplexy by orexin neurons, since activation of serotonergic neurons or activation of serotonergic nerve terminals in the amygdala prevents cataplexy (*Hasegawa et al., 2014*; *Hasegawa et al., 2017*). Here, we show that

**Table 2.** Comparison of parameters between DT sleep and cataplexy.

| Parameter | DT sleep | Cataplexy |
|---|---|---|
| Neuron dependence | Loss of both MCH and orexin neurons necessary | Only loss of orexin neurons required; loss of MCH neurons exacerbates |
| EMG amplitude | Low | Low |
| Delta EEG spectral power | High | Low |
| Theta EEG spectral power | High | High |
| Earliest detection | 1 week post-ablation | 2 weeks post-ablation |
| Progressive increase as neurodegeneration proceeds? | No | Yes |
| Time-of-day occurrence | Both light and dark phases | Primarily in dark phase in OX mice; both phases in OXMC mice |
| Occurrence | Spontaneous | Spontaneous |
| Mean bout duration | Similar during both light and dark phases:~15 s | Similar during both light and dark phases:~50 s in OX mice but ~ 100 s in OXMC mice |
| Response to tactile stimulation | High; immediate arousal | Low |
| Position in home cage during occurrence | Random | Random |
| Grooming more likely to precede occurrence | Yes | No |
| Running more likely to precede occurrence | No | Yes |
| Response to chocolate | No effect | Increased |
| Response to clomipramine | No effect | Decreased |
| Response to modafinil | Decreased | No effect |

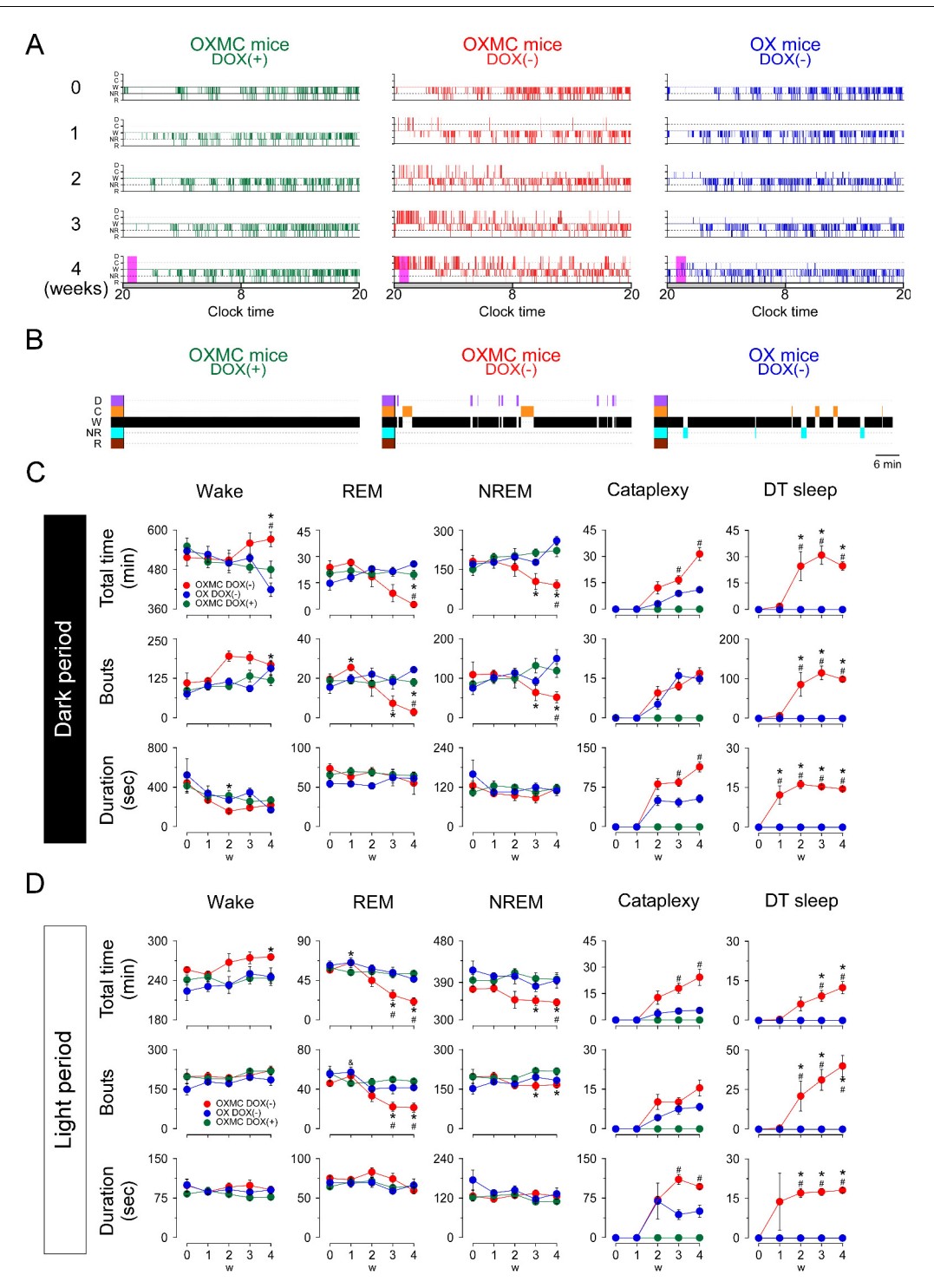

**Figure 5.** Effects of dual orexin- and MCH-neuron ablation on sleep/wake architecture in OXMC mice. (**A**) Hypnograms from OXMC DOX(+), OXMC DOX(-) and OX DOX(-) mice before (0 week) and during dual orexin and MCH neuron ablation (1–4 weeks). Clock time and light-dark phase are shown below the hypnograms. D, DT sleep; C, cataplexy; W, wake; NR, NREM sleep; R, REM sleep. (**B**) Hypnograms for the 1 hr period after dark onset indicated by magenta rectangles in the 4 weeks hypnogram in A. (**C**) Total duration (upper panels), number of bouts (middle) and mean bout duration (lower panels) for wake, NREM, REM, cataplexy and DT sleep during the dark phase. (**D**) As in C but for the light phase. Values are means ± SEM. (OXMC DOX(+): (n = 7), OXMC DOX(-): (n = 8), OX DOX(-): (n = 4)). *p<0.05 vs OXMC DOX(+). #p<0.05 vs OX DOX(-). Data were analyzed by one-way ANOVA followed by the Bonferroni *post hoc* test.

The online version of this article includes the following source data for figure 5:

**Source data 1.** Source data for *Figure 5A-D*.

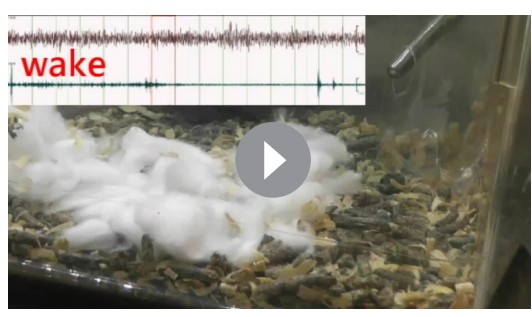

**Video 2.** Tactile stimulation during cataplexy in orexin- and MCH neuron-ablated mice.
https://elifesciences.org/articles/54275#video2

MCH neurons decrease cataplexy duration but not cataplexy bout frequency. This difference might suggest that the mechanism of MCH neurons to prevent cataplexy is different from that of serotonergic neurons.

Our in situ hybridization results as well as previous reports (*Mickelsen et al., 2017*; *Mickelsen et al., 2019*) suggest that both orexin and MCH neurons are glutamatergic neurons. Among LH glutamatergic neurons, these two types of neurons are a relatively minor population (the proportion of MCH neurons and orexin neurons was 15% and 24%, respectively). These results suggest that ablation of a small number of additional LH glutamatergic neurons (10–15%) that co-express MCH exacerbates the narcolepsy phenotype in orexin neuron-ablated mice, underscoring the important role of MCH neurons to suppress cataplexy.

Neurons in the amygdala innervate and suppress the ventrolateral periaqueductal grey (vlPAG/ LPT), DR, LC and tuberomammilary nucleus (TMN), areas that are known to be involved in the regulation of REM sleep or cataplexy (*Burgess et al., 2013*). It has been reported that MCH neurons also inhibit DR, LC, TMN and vlPAG neurons (*Torterolo et al., 2008*; *Sapin et al., 2010*; *Del Cid-Pellitero and Jones, 2012*; *Jego et al., 2013*). These areas might be part of the neural circuit contributing to cataplexy prevention since MCH neurons may innervate different types of neurons in these areas. Alternatively, other downstream targets of MCH neurons such as the sublaterodorsal tegmental nucleus might be involved in the regulation of cataplexy and REM sleep (*Boissard et al., 2003*; *Monti et al., 2016*). Recently, chemogenetic activation of MCH neurons in *Hcrt* knockout mice was shown to increase cataplexy bout duration without affecting the number of bouts (*Naganuma et al., 2018*), a finding that is inconsistent with our results. However, the absence of orexin during development in constitutive *Hcrt* knockout mice might induce compensatory changes in neural circuitry, since *Hcrt* knockout mice have many fewer cataplexy bouts per night than DTA mice in which orexin neurons are ablated after maturation (*Tabuchi et al., 2014*). In addition, the chemogenetic activation paradigm used by Naganuma et al. could have induced strong activation of MCH cells above the physiological range. Recently, we reported that MCH neurons can be divided in three types by activation pattern: wake-active, REM-active and wake- and REM-active (*Izawa et al., 2019*). Activation of wake-active MCH neurons within the physiological range in orexin neuron-ablated mice would likely be valuable to further understand the role of MCH neurons in cataplexy.

Orexin- and MCH-neurons double-ablated mice frequently displayed brief (~15 s) episodes of behavioral arrest during wakefulness that we called DT sleep, which was characterized by high power δ and θ waves in the EEG with low EMG amplitude. The first appearance of DT sleep was at 1 week DOX(-), which was sooner after DOX removal than the appearance of cataplexy at 2 weeks and suggests that DT sleep may be due to a smaller reduction in the number of orexin and MCH neurons. When initially observed, we assumed that DT sleep was cataplexy since both behaviors shared the characteristic of sudden behavioral arrest during wakefulness. However, the EEG spectrum was clearly distinct from cataplexy. Another critical difference between DT sleep and cataplexy was that mice responded to tactile stimulation by immediately returning to wakefulness from DT sleep but not from cataplexy. Furthermore, the behavioral arrest of DT sleep was observed without complete muscle atonia; thus, mice could maintain posture during DT sleep.

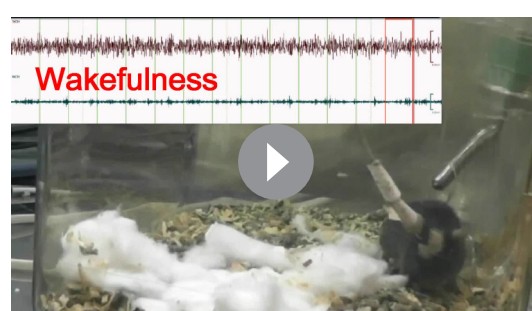

**Video 3.** Tactile stimulation during DT sleep in orexin- and MCH neuron-ablated mice.
https://elifesciences.org/articles/54275#video3

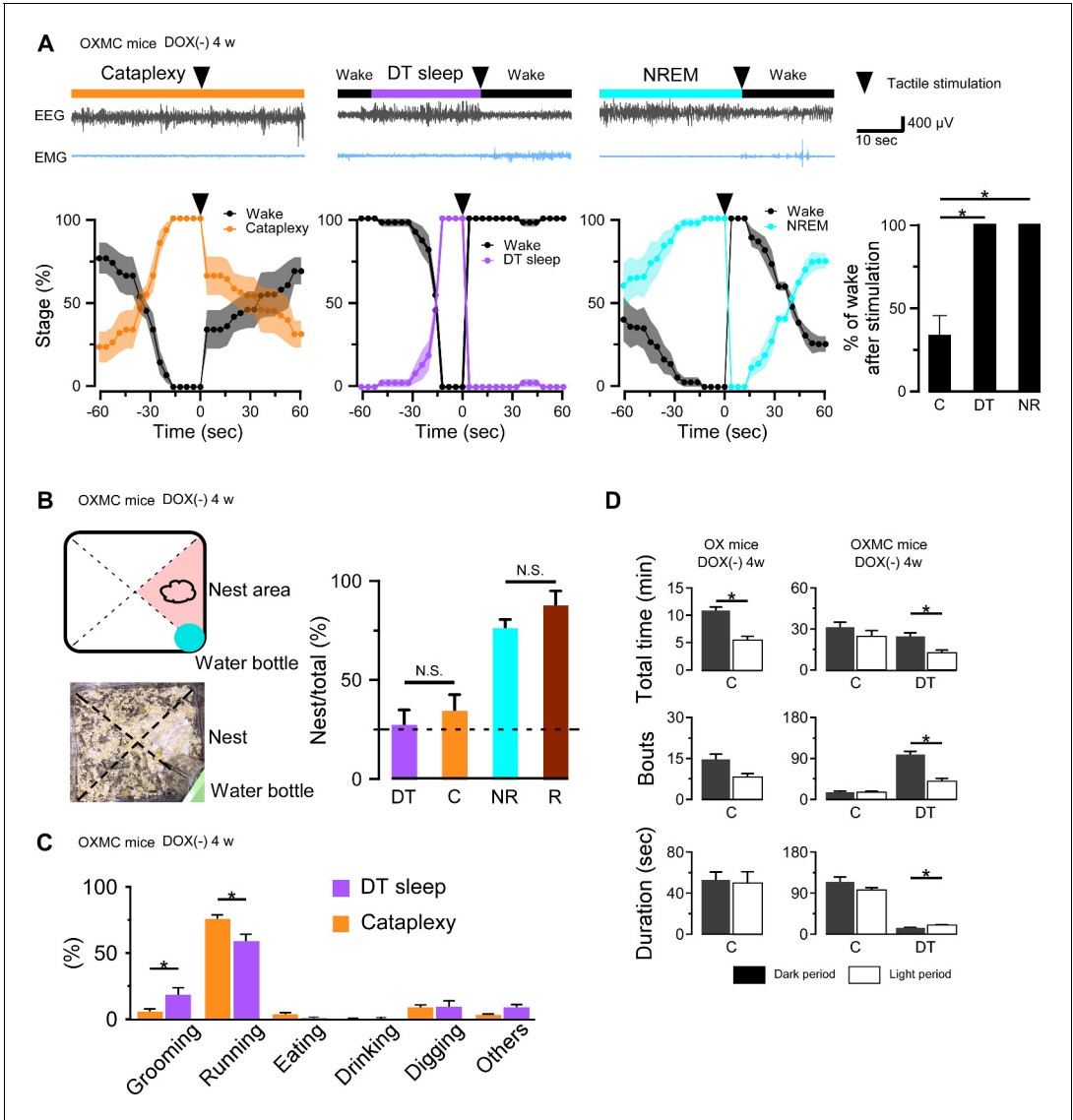

**Figure 6.** DT sleep is distinct from cataplexy. (**A**) Upper panels show representative traces of EEG and EMG when tactile stimulation (arrowheads) was performed during cataplexy, DT sleep and NREM sleep in OXMC DOX(-) mice after 4 weeks in the DOX(-) condition. Lower panels show the percentage of each state 1 min before and 1 min after tactile stimulation. Vigilance states were determined in 4 s epochs. Bar graphs shows the percentage of wakefulness during the first 3 epochs (12 s) after tactile stimulation during cataplexy, DT sleep (DT) and NREM sleep (NR). *p<0.05, one-way ANOVA followed by the Bonferroni *post hoc* test (OXMC DOX(-) (n = 6)). (**B**) Schematic drawing of home cage (upper left) showing location of nest and water bottle and corresponding picture (lower left). Bar graphs show the percentage of DT sleep (DT), cataplexy (C), NREM (NR), and REM sleep (R) occurring in the nest area in OXMC (-) mice. Dashed line in the bar graph indicates chance level (25%). *p<0.05, one-way ANOVA followed by the Bonferroni *post hoc* test (OXMC DOX(-) (n = 7)). (**C**) Bar graphs indicate the behaviors observed before cataplexy and DT sleep. Values are mean ± SEM. *p<0.05, paired t-test (OXMC DOX(-) (n = 7)). (**D**) Total time, number of bouts and mean bout duration of cataplexy and DT sleep during the dark and light phases in OX DOX(-) (n = 4) mice (left) and OXMC DOX(-) (n = 7) mice (right). *p<0.05, paired t-test.
The online version of this article includes the following source data for figure 6:

**Source data 1.** Source data for *Figure 6A-D*.

This observation underscores the difference between DT sleep and cataplexy, since muscle atonia is an indispensable component of cataplexy.

Pharmacological assessments further confirmed differences between DT sleep and cataplexy. Chocolate is known to increase cataplexy by activating the prefrontal cortex and amygdala (*Burgess et al., 2013*). Chocolate increased cataplexy in double-ablated mice, but did not increase DT sleep. Clomipramine is a selective serotonin reuptake inhibitor that can inhibit REM sleep and

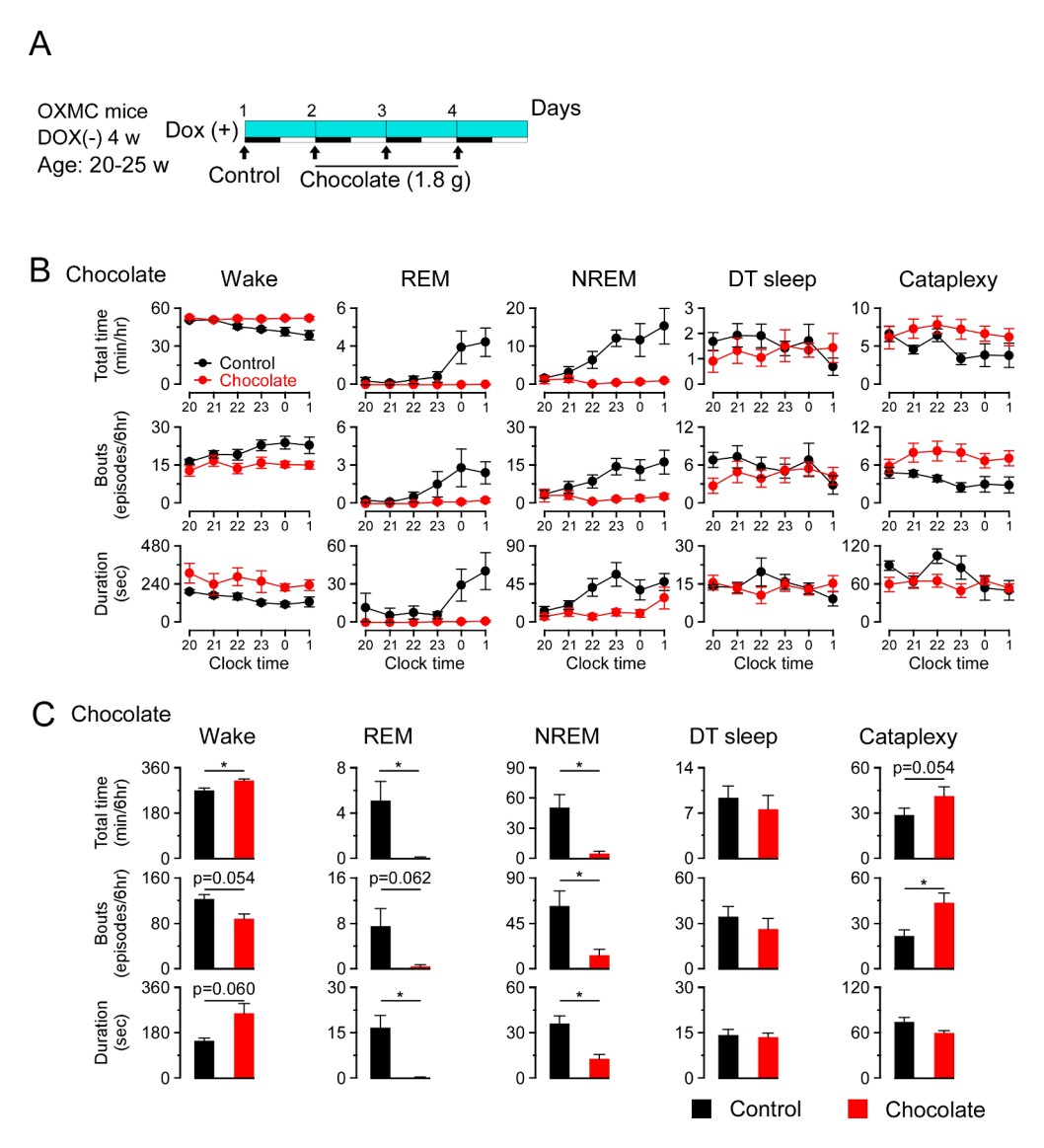

**Figure 7.** Effects of chocolate availability on DT sleep and cataplexy in OXMC mice. (**A**) Schematic showing the experimental protocol of chocolate availability in OXMC DOX(-) mice (n = 8). To avoid further neuron degeneration during the test, DOX chow was re-introduced after 4 weeks in the DOX(-) condition; light blue bar indicates this DOX(+) condition. Light-dark phases are indicated as white and black bars. Chocolate was made available for 15 min prior to light offset (19:45-20:00). (**B**) Hourly amounts of time, number of bouts and mean bout duration for each vigilance state during the first 6 hr of the dark phase after chocolate administration. Values are mean ± SEM. (**C**) Total time, number of bouts and mean bout duration for each vigilance state during the first 6 hr of the dark phase after chocolate administration. *p<0.05, vs control or vehicle. Values are mean ± SEM. Data were analyzed by paired t-test.

The online version of this article includes the following source data for figure 7:

**Source data 1.** Source data for *Figure 7B and C*.

cataplexy (*Willie et al., 2003*). Clomipramine significantly decreased cataplexy in double-ablated mice, but did not affect the number of bouts or mean bout duration of DT sleep. Modafinil is known to be a dopamine reuptake inhibitor that can promote wakefulness (*Willie et al., 2005*). Although modafinil significantly increased wakefulness, decreased NREM sleep and decreased the duration of DT sleep, the number of DT bouts was unaffected. Based on these results, DT sleep is classified as sleep but is distinct from NREM or REM sleep. Modafinil decreased the duration of DT sleep and DT

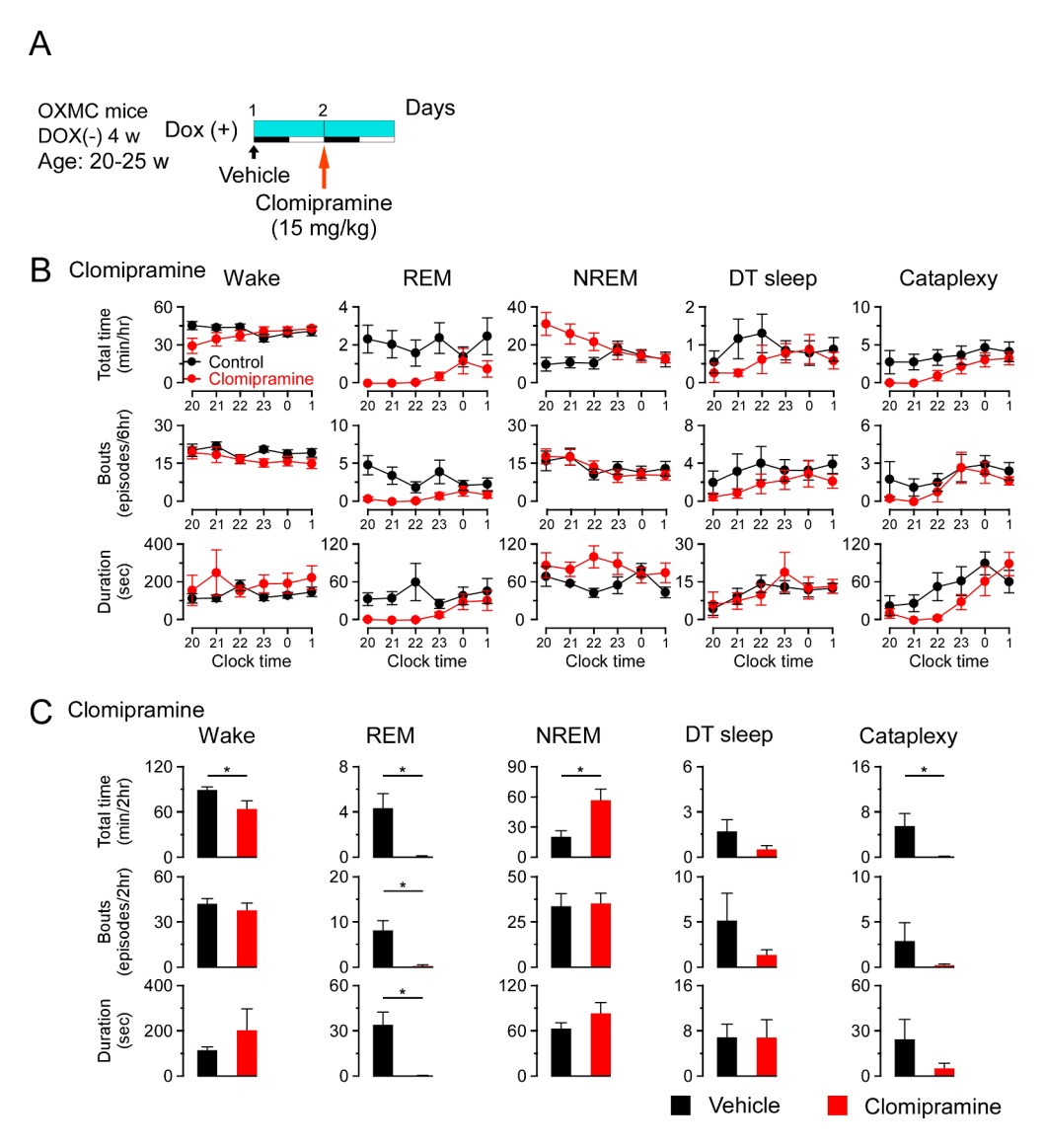

**Figure 8.** Effects of clomipramine on DT sleep and cataplexy in OXMC DOX(-) mice. (**A**) Schematic showing the experimental protocol for clomipramine administration (15 mg/kg, i.p.) in OXMC DOX(-) mice (n = 8). To avoid further neuron degeneration during the test, DOX chow was re-introduced after 4 weeks in the DOX(-) condition; light blue bar indicates this DOX(+) condition. Light-dark phases are indicated as white and black bars. Clomipramine or vehicle was administered by i.p. injection prior to dark onset (19:45-20:00) as indicated by the arrows. (**B**) Hourly amounts of time, number of bouts and mean bout duration for each vigilance state during the first 6 hr of the dark phase after clomipramine administration. Values are mean ± SEM. (**C**) Total time, number of bouts and mean bout duration for each vigilance state during the first 2 hr of the dark phase after clomipramine administration. *p<0.05, vs control or vehicle. Values are mean ± SEM. Data were analyzed by paired t-test.

The online version of this article includes the following source data for figure 8:

**Source data 1.** Source data for *Figure 8B and C*.

sleep was often observed after grooming behavior, suggesting that dopaminergic neurons might be involved in the initiation of DT sleep since dopaminergic neurons are activated during grooming (*Fornaguera et al., 1995*; *Cromwell et al., 1998*).

The transition from NREM to REM sleep is well known to be characterized by relatively high power in δ and θ waves (*Benington et al., 1994*). Thus, we compared DT sleep to the transition from NREM to REM sleep. We found that δ, θ, α and β wave power were not significantly different

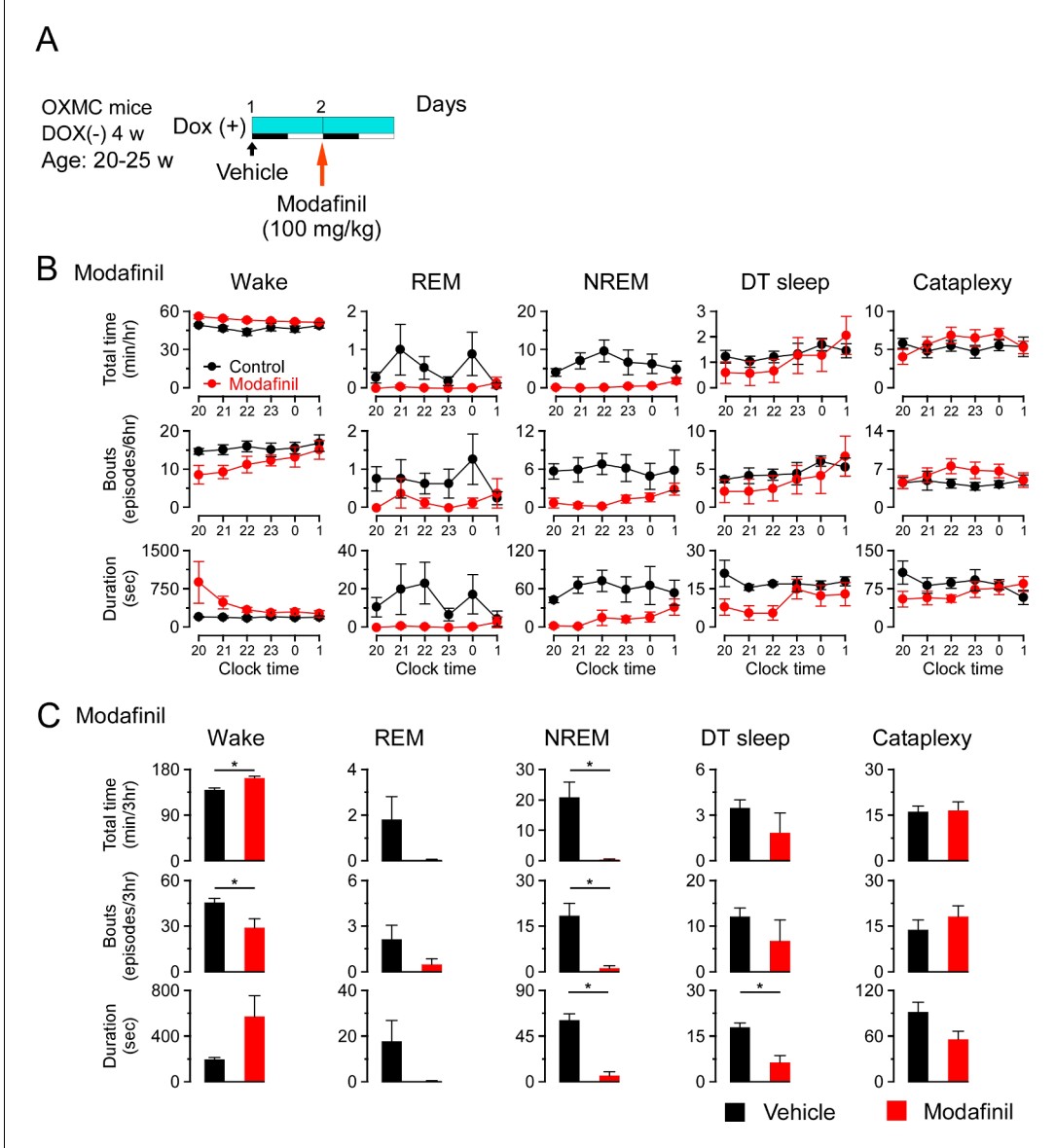

**Figure 9.** Effects of modafinil on DT sleep and cataplexy in OXMC DOX(-) mice. (**A**) Schematic showing the experimental protocol for modafinil administration (100 mg/kg, i.p.) in OXMC DOX(-) mice (n = 8). To avoid further neuron degeneration during the test, DOX chow was re-introduced after 4 weeks in the DOX(-) condition; light blue bar indicates this DOX(+) condition. Light-dark phases are indicated as white and black bars. Modafinil or vehicle was administered by i.p. injection prior to dark onset (19:45-20:00) as indicated by the arrows. (**B**) Hourly amounts of time, number of bouts and mean bout duration for each vigilance state during the first 6 hr of the dark phase after modafinil administration. Values are mean ± SEM. (**C**) Total time, number of bouts and mean bout duration for each vigilance state during the first 3 hr of the dark phase after modafinil administration. *p<0.05, vs control or vehicle. Values are mean ± SEM. Data were analyzed by paired t-test.

The online version of this article includes the following source data for figure 9:

**Source data 1.** Source data for *Figure 9B and C*.

between DT sleep and NREM to REM sleep transitions. The activity of orexin neurons is high in wakefulness and low in NREM and REM sleep (*Lee et al., 2005*). In comparison, the activity of MCH neurons is high in REM sleep and low in NREM sleep and wakefulness (*Hassani et al., 2009*; *Blanco-*

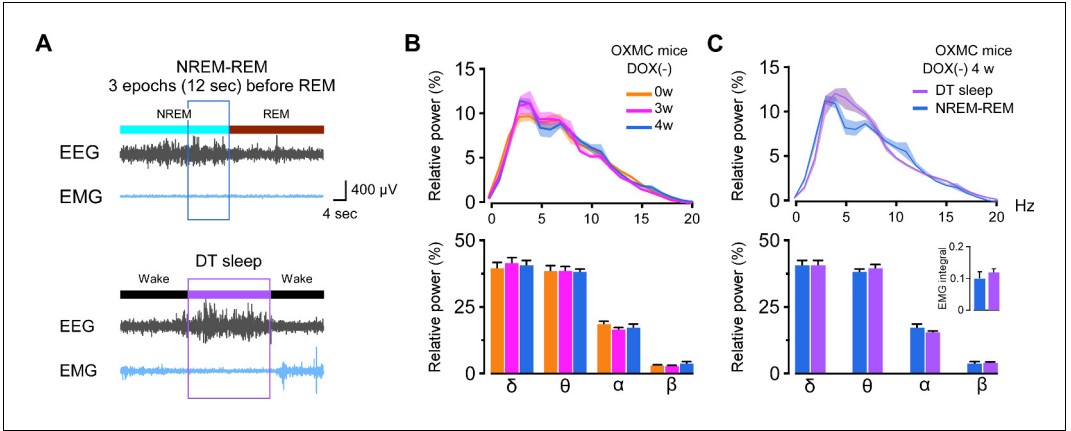

**Figure 10.** Comparison of EEG spectral power during DT sleep to the transition from NREM to REM sleep. (**A**) Typical traces of EEG and EMG during the transition from NREM to REM sleep (upper) and during DT sleep (lower). Three epochs during the transition from NREM sleep to REM sleep were analyzed. (**B**) Comparison of relative EEG power during the transition from NREM to REM sleep in OXMC DOX(-) mice during ablation (0, 3, and 4 weeks DOX(-)). The upper graph shows relative EEG power distribution. The lower bar graph shows the relative EEG power in each spectral band (δ, θ, α and β). Values are mean ± SEM. Data were analyzed by one-way ANOVA followed by the Bonferroni *post hoc* test. (**C**) Comparison of relative EEG power and EMG integral (inset) during DT sleep and the transition from NREM to REM sleep in OXMC DOX(-) mice. Upper graph shows relative EEG power distribution; lower bar graph shows relative EEG power in each band (δ, θ, α and β). Values are mean ± SEM. Paired t-test.

The online version of this article includes the following source data for figure 10:

**Source data 1.** Source data for *Figure 10B and C*.

*Centurion et al., 2019*; *Izawa et al., 2019*). Therefore, the minimum activity of these two types of neurons should occur just before initiation of REM sleep. It is possible that this transition might be manifest as DT sleep, since the brains of dual orexin- and MCH neuron-ablated mice are similar to conditions in which both orexin neurons and MCH neurons are inactive. Furthermore, NREM and REM sleep were attenuated in double-ablated mice, suggesting that DT sleep enables a compensatory role of both NREM and REM sleep. Whereas the loss of orexin neurons in narcolepsy has been proposed to disrupt a 'flip/flop switch' that facilitates the normal transition from wakefulness to sleep (*Saper et al., 2001*), the dual loss of orexin and MCH neurons appears to prolong the normal transition from NREM to REM sleep through the DT sleep state.

To date, specific degeneration of MCH neurons has not been reported in humans. Ablation of MCH neurons in mice induces a relatively weaker effect on sleep and wakefulness compared with ablation of orexin neurons (*Tabuchi et al., 2014*; *Tsunematsu et al., 2014*). MCH concentration in cerebrospinal fluid is not typically measured in sleep disorders patients. However, a patient with hypersomnia had low concentrations of both orexin (95 pg/ml) and MCH peptide (6 pg/ml) in the cerebrospinal fluid (Wada et al., Japan sleep annual meeting 2018, P-128). Detailed study of this and similar cases will help to understand the role of MCH neurons on sleep/wakefulness in humans.

Here, we revealed a new role of MCH neurons to prevent cataplexy and regulate sleep and wakefulness. Orexin neurons and MCH neurons interact in the hypothalamus and potentially in terminal projection sites to influence sleep/wake control. Since human studies have already suggested interaction between these two systems (*Blouin et al., 2013*), further investigations are warranted to reveal the regulatory mechanisms underlying the roles of these two systems in the control of sleep/wakefulness and cataplexy.

# Materials and methods

### Key resources table

| Reagent type (species) or resource | Designation | Source or reference | Identifiers | Additional information |
|---|---|---|---|---|
| Genetic reagent (*Mus musculus*) | *Hcrt-tTA* | **Tabuchi et al., 2014** | RRID:IMSR_APB:7778 | |
| Genetic reagent (*Mus musculus*) | *Pmch-tTA* | **Tsunematsu et al., 2014** | RRID:IMSR_RBRC05844 | |
| Genetic reagent (*Mus musculus*) | *TetO DTA* | Jackson lab | RRID:IMSR_JAX:008468 | |
| Genetic reagent (*Mus musculus*) | *Hcrt-tTA;TetO DTA* | **Tabuchi et al., 2014** | RRID:MGI:5583048 | |
| Genetic reagent (*Mus musculus*) | *Hcrt-tTA; Pmch-tTA; TetO DTA* | This paper | | Mating of the *Hcrt-tTA; TetO DTA* with the *Pmch-tTA* mice |
| Antibody | Rabbit anti-Pmch antibody | Sigma-Aldrich | RRID:AB_260690 | (1:1000) |
| Antibody | Goat polyclonal anti-Hcrt | Santa Cruz Biotechnology | RRID:AB_653610 | (1:1000) |
| Antibody | Mouse anti-tryptophan hydroxylase antibody | Sigma-Aldrich | RRID:AB_261587 | (1:500) |
| Antibody | Rabbit anti-tyrosine hydroxylase antibody | Merck | RRID:AB_390204 | (1:1000) |
| Antibody | Biotinylated horse anti-goat IgG antibody | Vector Laboratories | RRID:AB_2336123 | (1:1000) |
| Antibody | Biotinylated goat anti-rabbit/mouse IgG antibody | Vector Laboratories | RRID:AB_2313606 | (1:1000) |
| Sequence-based reagent | *Hcrt* | ACD Bio | Cat No. 490461-C2; RRID:SCR_012481 | (1:1500) |
| Sequence-based reagent | *Pmch* | ACD Bio | Cat No. 478721-C1; RRID:SCR_012481 | (1:1500) |
| Sequence-based reagent | *Slc32a1* (vGAT) | ACD Bio | Cat No. 319191-C4; RRID:SCR_012481 | (1:750) |
| Sequence-based reagent | *Slc17a6* (vGlut2) | ACD Bio | Cat No. 319171-C3; RRID:SCR_012481 | (1:750) |
| Chemical compound, drug | Opal520 | PerkinElmer | RRID:SCR_012163 | (1:1000) |
| Chemical compound, drug | Opal620 | PerkinElmer | RRID:SCR_012163 | (1:1000) |
| Chemical compound, drug | Chocolate | Meiji | Milk chocolate | (1.7–1.9 g/once) |
| Chemical compound, drug | Clomipramine | Sigma-Aldrich | C7291-1G; RRID:SCR_008988 | (15 mg/kg) |
| Chemical compound, drug | Modafinil | Cephalon Inc (Teva Pharmaceutical Industries Ltd.) | | (100 mg/kg) |
| Software, algorithm | Origin 2017 | Lightstone | RRID:SCR_014212; version 2017 | |

*Continued on next page*

*Continued*

| Reagent type (species) or resource | Designation | Source or reference | Identifiers | Additional information |
|---|---|---|---|---|
| Software, algorithm | SleepSign | Kissei Comtec | | |
| Software, algorithm | ImageJ | https://imagej.nih.gov/ij/ | RRID:SCR_002285 | |

## Experimental procedures

### Animal usage

All experimental procedures were performed in accordance with the guide of the Institutional Animal Care and Use Committes at the Research Institute of Environmental Medicine at Nagoya University and SRI International. All efforts were made to decrease animal suffering and to minimize the number of animals used.

### Animals

All male mice had a C57BL/6J background and were housed on a 12 hr dark-light cycle (light on clock time: 8:00-20:00). The room temperature (RT) was kept at 25 ± 2°C, and humidity was maintained between 40% to 60%. The light intensity was 200 to 300 lux. Food and water were provided ad libitum. *Hcrt-tTA* mice express the mammalianized tetracycline-controlled transcriptional activator (tTA) in orexin neurons controlled by human *Hcrt* promoter (*Sakurai et al., 1999*; *Tabuchi et al., 2014*). *Pmch-tTA* mice express tTA in MCH neurons controlled by *Pmch* promoter (*Tsunematsu et al., 2014*). *TetO DTA* mice express diphtheria toxinA (DTA) that is under control of the Tet-off system (B6.Cg-Tg (*TetO DTA*) 1Gfi/j, The Jackson Laboratory, USA). The tTA protein binds to the tetracycline operator (TetO) and promotes expression of DTA. Doxycycline (DOX)-containing chow (DOX chow) was produced by mixing 10% DOX powder (Kyoritus Seiyaku, Japan) with normal chow (Labo MR Stock, Nosan, Japan) at the final concentration of 100 mg/kg. *Hcrt-tTA* (Tg/-); *TetO DTA* (Tg/-); *Pmch-tTA* (Tg/-) mice (OXMC) expressed tTA exclusively in both orexin and MCH neurons. Removing DOX-containing chow resulted in ablation of orexin and MCH neurons (*Tabuchi et al., 2014*; *Tsunematsu et al., 2014*). Mating mice were fed with DOX chow for at least 1 week before the start of mating. During the prenatal and early postnatal periods, DOX was supplied via maternal circulation or lactation. After weaning, mice were fed with DOX chow until the day of the experiment. Genotypes of mice were determined by polymerase chain reaction (PCR) and electrophoresis. Three pairs of primers were used to check the genotypes: *Hcrt-tTA* (5'-AAGTCGACGG TGTCTGGCGCTCAGGGTG-3', 5'-GCAGCGGCCATTCCTTGG-3'), *Pmch-tTA (5'-CCAGGGTCTCG TACTGCTTC-3', 5'-AAGCATCAAACTAAGGCCAC-3')*, and *TetO DTA* (5'-GGCATTATCCACTTTTAG TGC-3', 5'-AGCAGAGCTCGTTTAGTGAACCGT-3').

### EEG and EMG surgical procedure

Male mice were anesthetized with 2% isoflurane (095–06573, FUJIFILM Wako Pure Chemical Corporation, Japan) and implanted with three EEG electrodes (U-1430–01, Wilco, Japan) on the skull and two EMG electrodes (AS633, Cooner wire, Mexico) in the rhomboid muscle at 10 weeks of age as previously described *Tabuchi et al. (2014)*; *Figure 1C*). Carprofen (Zoetis Inc, Japan), 20 mg/kg (subcutaneous injection), was administered the day of, and the day after, surgery for its anti-inflammatory and analgesic properties. After surgery, mice were housed separately for about 7 days for recovery. A cable with a slip ring (Kissei Comtec Co., Ltd, Japan) was connected to mice in the cage for 7 days before the initiation of EEG and EMG recordings.

### Sleep/wake recordings

EEG and EMG signals were filtered at EEG 1.5–30 Hz and at EMG 15–300 Hz and amplified by an amplifier (AB-610J, Nihon Koden, Japan). The digital sampling rate was 128 Hz. Animal behavior was monitored through a Charged Coupled Device (CCD) video camera (SPK-E700CHP1, Keiyo Techno, Japan) and an infrared activity sensor (Kissei Comtec). All EEG and EMG data were recorded by VitalRecorder (Kissei Comtec) and analyzed by SleepSign software (Kissei Comtec).

## Vigilance state analysis

EEG analysis was performed by Fast Fourier Transform (FFT). Power spectra profiles over a 0–10 Hz window with 1 Hz resolution for $\delta$ ($1 \leq \delta <6$ Hz) and $\theta$ ($6 \leq \theta <11$ Hz) bandwidths were calculated. EEG and EMG were automatically screened in 4 s epochs by SleepSign and classified as previously reported (*Tsunematsu et al., 2011*; *Tabuchi et al., 2014*; *Tsunematsu et al., 2014*). Wake was characterized by high EMG amplitude or locomotion score with low EEG amplitude, NREM sleep was characterized by low EMG amplitude and high EEG $\delta$ power, and REM sleep was characterized by low EMG and low EEG amplitude with over 50% of $\theta$ activity. Cataplexy was defined by muscle atonia lasting more than 10 s after more than 40 s of wakefulness, and low amplitude with over 50% of $\theta$ activity in the EEG (*Scammell et al., 2009*). DT sleep was defined by the following criteria: (1) a period of behavioral arrest, (2) that was preceded by at least 40 s of wakefulness, (3) characterized by high $\delta$ and $\theta$ power (at least 60–80% of the $\delta$ power found in NREM sleep) in the EEG, (4) low EMG activity, and (5) was terminated by a return to wakefulnes. Vigilance state classifications were automatically assigned by SleepSign software and then visually corrected. The transition from NREM to REM sleep was identified as the three epochs (12 s) before a REM sleep bout. EEG power spectral density were analyzed and viewed using the Signal Processing Toolbox of Matlab (Mathworks, USA).

## Behavioral assessment

Behaviors were manually identified from video recordings. Behaviors occurring during the 4 s epoch immediately preceding DT sleep or cataplexy bouts were analyzed and calculated as percentages of the total recording time.

## Tactile stimulation

Tactile stimulation was performed by applying a gentle touch around the face using a brush during the light phase (11:00-18:00) (*Videos 2* and *3*). Six OXMC DOX(-) mice (20–26 weeks old) were studied. Stimulations were conducted when immobility had lasted at least 10 s during NREM sleep, cataplexy or DT sleep. The interval between individual stimulations was at least 1 min. Vigilance states that occurred after stimulation were identified.

## Identification of the location of each sleep/wake stage within the home cage

We divided each home cage into four quadrants, one of which contained a nest (*Figure 6B*). The home cage quadrant in which each mouse was located at the beginning of each vigilance state was identified by video recording.

## Pharmacological treatments and chocolate application

All administrations were performed before light offset (19:45-20:00). Chocolate (1.7–1.9 g, milk chocolate, Meiji) was placed into the cage and mice could freely access chow and water at the same time. Clomipramine (15 mg/kg) and modafinil (100 mg/kg) were administered by intraperitoneal (i.p.) injection. Clomipramine hydrochloride (C7291-1G, MilliporeSigma, USA) was dissolved in saline (035–081517, Otsuka Pharmaceutical Co., Ltd, Japan). Modafinil (Cephalon Inc, USA) was dissolved in 5% dimethyl sulfoxide (07-4875-5, MilliporeSigma) and the mixed solution was diluted in saline.

## Immunohistochemistry

Mice were deeply anesthetized with an i.p. injection of somnopentyl (6.48 mg/kg) (Kyoritsu Seiyaku Corporation, Tokyo, Japan) and 3% isoflurane, and then sequentially perfused with saline and 10% formalin (066–03847, FUJIFILM Wako). The brains were removed and immersed in 10% formalin overnight at 4°C and then immersed in a 30% sucrose solution in PBS for at least 2 d. The brains were frozen in embedding solution (4583, Sakura Finetek Japan, Japan) and stored in a −80°C freezer. Brains were subsequently sectioned at 40 μm thickness in a cryostat (CM3050-S, Leica Microsystems K.K., Japan).

Coronal sections of mouse brains were placed in phosphate buffer (PBS) with 0.3% $H_2O_2$ (H1009-500ML, MilliporeSigma) to inactivate endogenous peroxidase for 40–45 min at RT. After washing three times for 10 min in PBS containing 0.25% Triton X-100 (35501–15, Nacalai Tesque, Japan) and

1% bovine serum albumin (A7905-500G, MilliporeSigma) (PBS-BX), brain sections were incubated in PBS-BX with rabbit anti-MCH antibody (1:1000, M8440, MilliporeSigma), goat anti-orexin-A antibody (1:1000, sc-8070, Santa Cruz, Dallas, USA), mouse anti-tryptophan hydroxylase antibody (1:500, T0678-100UL, MilliporeSigma), or rabbit anti-tyrosine hydroxylase antibody (1:1000, AB152, Merck, Germany) overnight at 4℃. Sections were washed 3 times for 10 min in PBS-BX and incubated with biotinylated horse anti-goat IgG antibody (1:1000, BA-9500, Vector Laboratories, USA) or biotiny-lated goat anti-rabbit/mouse IgG antibody (1:1000, BA-1000, Vector Laboratories) in PBS-BX for 2 hr at RT. Sections were then washed in PBS-BX 3 times for 10 min and reacted with avidin-biotin perox-idase complex (PK-6100, Vectastain, USA) dissolved in PBS-BX at RT for 30 min. After washing 3 times for 10 min in PBS-BX, bound-peroxidase was visualized by DAB-buffer tablets (1.02924.0001, Merck) in distilled water with 0.0015% $H_2O_2$ resulting in a golden-brown reaction product. For two-color staining, the process described previously was performed twice. The 2nd staining process used the DAB solution mixed with 2.5% ammonium nickel (II) (140–01015, FUJIFILM Wako), resulting in a black reaction product.

After staining, sections were mounted and dried at RT. Sections were sequentially rinsed in 70%, 90%, 100% (twice, 057–00451, FUJIFILM Wako) ethanol and xylene (twice, 244–00081, FUJIFILM Wako) for 30 s each. Brain sections were then mounted by Entellan (1.07961.0100, Merck) and dried at RT. Photomicrographs were obtained using a microscope (BZ-9000, Keyence, Japan). Every fourth section in the LH was used for cell counting (ImageJ).

## Statistical analyses

Data were analyzed by paired or unpaired *t*-test using Excel (Microsoft, USA) or one-way Analysis of Variance (ANOVA) (with/without repeated measures) followed by Bonferroni's *post hoc* test using OriginPro 2017 software (ORIGIN 2017 Graphing and Analysis, OriginLab, USA).

## In situ RNA hybridization by RNAscope

Mouse brains were isolated as described in the Immunohistochemistry section above. After freezing in embedding solution at −80℃, embedded brains were sectioned at 25 μm thickness and mounted on glass slides. The slides were then treated with RNAscope multiplex fluorescent v2 (#323100, Advanced Cell Diagnostics, Hayward, CA) according to the RNAscope standard protocol. Briefly, slides were incubated with hydrogen peroxide at RT for 10 min, followed by boiling with target retrieval reagent at 98 ~ 102℃ for 5 min, and protease digestion (Protease III) at 40℃ in an HybEZ hybridization oven (Advanced Cell Diagnostics) for 30 min. Subsequently, slides were incubated at 40℃ with target probes for 2 hr and slides were then washed in wash buffer (WB) twice (2 min each) at room temperature. Sections were then incubated in AMP1 buffer for 30 min, AMP2 buffer for 30 min, and AMP3 buffer for 15 min. After each hybridization step, slides were washed twice in WB. For fluorescent detection, the RNA probes were conjugated to Opal 520 or 620 with the TSA Plus Fluo-rescence system (Perkin Elmer, Waltham, MA). Slices were mounted with ProLong Gold Antifade Mountant (Thermo Fisher Scientific, Waltham, MA). One fourth of brain slices were randomly chosen from 5 mice for four in situ hybridization combinations, as illustrated in *Figure 1—figure supplement 1*. A dotted circle (diameter: 1.2 mm) was located within the core LH region where orexin and MCH neurons are at a high density and the number of neurons were counted inside the circle as shown in *Figure 1—figure supplement 1*. The zona incerta, medial globus pallidus, and ventrome-dial hypothalamus were excluded from counting.

## Acknowledgements

We thank S Tsukamoto, A Inui, Y Miyoshi, E Imoto, S Nasu and N Fukatsu for technical assistance. This work was supported by JST CREST (JPMJCR1656) to AY, KAKENHI grants (26293046, 26640041, 16H01271, 17H05563, 18H05124, 18KK0223 and 18H02523) to AY, KAKENHI grant (18H02477) to DO and NIH R01 NS098813 to TSK.

# Additional information

## Funding

| Funder | Grant reference number | Author |
| --- | --- | --- |
| Ministry of Education, Culture, Sports, Science and Technology | 26293046 | Akihiro Yamanaka |
| Japan Science and Technology Agency | JPMJCR1656 | Akihiro Yamanaka |
| NIH Blueprint for Neuroscience Research | R01 NS098813 | Thomas S. Kilduff |
| Ministry of Education, Culture, Sports, Science and Technology | 26640041 | Akihiro Yamanaka |
| Ministry of Education, Culture, Sports, Science and Technology | 16H01271 | Akihiro Yamanaka |
| Ministry of Education, Culture, Sports, Science and Technology | 17H05563 | Akihiro Yamanaka |
| Ministry of Education, Culture, Sports, Science and Technology | 18H05124 | Akihiro Yamanaka |
| Ministry of Education, Culture, Sports, Science and Technology | 18KK0223 | Akihiro Yamanaka |
| Ministry of Education, Culture, Sports, Science and Technology | 18H02523 | Akihiro Yamanaka |
| Ministry of Education, Culture, Sports, Science and Technology | 18H02477 | Daisuke Ono |

The funders had no role in study design, data collection and interpretation, or the decision to submit the work for publication.

## Author contributions

Chi Jung Hung, Data curation, Formal analysis, Writing - original draft; Daisuke Ono, Formal analysis, Supervision; Thomas S Kilduff, Formal analysis, Supervision, Funding acquisition, Writing - review and editing; Akihiro Yamanaka, Conceptualization, Resources, Supervision, Funding acquisition, Writing - review and editing

## Author ORCIDs

Chi Jung Hung (iD) https://orcid.org/0000-0002-1169-7953
Daisuke Ono (iD) https://orcid.org/0000-0001-5967-2002
Thomas S Kilduff (iD) https://orcid.org/0000-0002-6823-0094
Akihiro Yamanaka (iD) https://orcid.org/0000-0001-6099-7306

## Decision letter and Author response

Decision letter https://doi.org/10.7554/eLife.54275.sa1
Author response https://doi.org/10.7554/eLife.54275.sa2

## Additional files

### Data availability

All data generated or analysed during this study are included in the manuscript and supporting files. Source data files have been provided for Figures 1-10 and Figure 1—figure supplement 1.

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
