## [Decision Letter]

**Acceptance summary:**

The mouse model with effective ablation of both orexin and MCH cells provide a powerful tool for studying the interaction between the two systems in sleep control. We found the behavioral characterization thorough and convincing. The study represents significant contribution to understanding normal and impaired sleep control.

**Decision letter after peer review:**

Thank you for submitting your article "Dual Orexin and MCH neuron-ablated mice display severe sleep attacks and cataplexy" for consideration by *eLife*. Your article has been reviewed by three peer reviewers, including Yang Dan as the Reviewing Editor and Reviewer #1, and the evaluation has been overseen by Laura Colgin as the Senior Editor. The following individuals involved in review of your submission have agreed to reveal their identity: Shinjae Chung (Reviewer #2).

The reviewers have discussed the reviews with one another and the Reviewing Editor has drafted this decision to help you prepare a revised submission.

Previous studies showed that killing orexin neurons decreases wakefulness and causes narcolepsy, and killing MCH neurons increases wakefulness. But how these two closely located neuronal population interact was unclear. Hung et al., ablate MCH, Orexin/Hypocretin or both to understand how these neuropeptide systems contribute to sleep and wake regulation. They performed thorough analyses to examine their sleep/wake patterns, and found that the double-ablated mice show increased wakefulness and decreased REM and NREM sleep, but cataplexy bout duration and total time also increased compared to orexin neuron-ablation only, suggesting that MCH neurons also suppress cataplexy. These results reveal previously unknown interactions between orexin and MCH systems.

The mice they generated allow very effective ablation of both cell types, and they provide a powerful tool for studying the interaction between the two systems. The behavioral characterization is thorough, and the data are very convincing. The distinction between DT sleep, normal sleep and cataplexy is quite useful, and the nest analysis in Figure 6 is particularly informative.

Essential revisions:

- Is there reason to believe that narcoleptics or any other sleep related neurological syndrome is linked to MCH cell loss? Why doesn't loss of MCH neurons alone cause cataplexy? The authors state: "However, it is possible that both orexin and MCH neurons degenerate in some narcolepsy patients with severe cataplexy or hypersomnia." Anything "is possible" but they have no evidence for this. It seems that this analysis could easily be done on existing CSF samples available in both Japan and the US. Why not do this before publication? The authors should at least add more discussions on these points.

- Number of mice used in Figure 7, Figure 8 and Figure 9 is not shown.

- Figure 4. According to the figure legend, Figure 3A shows the "Relative EEG power in three strains of mice in the DOX(+) condition during the dark phase." Similarly, the main text (subsection 'Dual ablated mice frequently displayed short behavioral arrests with high spectral power in the δ and θ bands of the EEG during wakefulness') also mentions three groups tested "in DOX(+) condition". But the legend in the Figure 4A (on the upper right) shows OXMC mice DOX(-), OXMC mice DOX(+), OX mice DOX(-). I suspect that the figure legend is correct?

- Figure 4D. For better comparison, could the authors include a comparison of the EMG amplitude during the different states?

- Figure 10. For better comparison, could the authors also compare the EMG amplitudes during DT sleep and NREM to REM transitions?

- The finding that MCH neurons also suppress cataplexy is interesting and surprising. One question is whether this is specific to MCH neurons or general to other GABAergic neurons in the same region. I am not sure if it's feasible to add a control experiment by ablating a similar number of neurons in the lateral hypothalamus that are not necessarily MCH neurons. If not, the authors can perhaps add a discussion of this point.

---

## [Author Response]

Previous studies showed that killing orexin neurons decreases wakefulness and causes narcolepsy, and killing MCH neurons increases wakefulness. But how these two closely located neuronal population interact was unclear. Hung et al., ablate MCH, Orexin/Hypocretin or both to understand how these neuropeptide systems contribute to sleep and wake regulation. They performed thorough analyses to examine their sleep/wake patterns, and found that the double-ablated mice show increased wakefulness and decreased REM and NREM sleep, but cataplexy bout duration and total time also increased compared to orexin neuron-ablation only, suggesting that MCH neurons also suppress cataplexy. These results reveal previously unknown interactions between orexin and MCH systems.The mice they generated allow very effective ablation of both cell types, and they provide a powerful tool for studying the interaction between the two systems. The behavioral characterization is thorough, and the data are very convincing. The distinction between DT sleep, normal sleep and cataplexy is quite useful, and the nest analysis in Figure 6 is particularly informative.Essential revisions:- Is there reason to believe that narcoleptics or any other sleep related neurological syndrome is linked to MCH cell loss? Why doesn't loss of MCH neurons alone cause cataplexy? The authors state: "However, it is possible that both orexin and MCH neurons degenerate in some narcolepsy patients with severe cataplexy or hypersomnia." Anything "is possible" but they have no evidence for this. It seems that this analysis could easily be done on existing CSF samples available in both Japan and the US. Why not do this before publication? The authors should at least add more discussions on these points.

We have previously shown that ablation of MCH neurons alone does not induce cataplexy-like behavioral arrests but affects both sleep/wakefulness and memory in mice. Specifically, MCH neuron-ablated mice increased wakefulness and decreased NREM sleep without affecting REM sleep amount (Tabuchi et al., 2014) and also improved retention hippocampus-dependent memories (Izawa et al., 2019). These facts suggest that MCH neurons have multiple roles in the regulation of sleep/wakefulness and memory.

Regarding the possibility of humans with degeneration of both orexin and MCH neurons: this suggestion is based on a collaboration with clinical investigators that is beyond the scope of the present study. Although the results have not yet been published, a patient with hypersomnia that had low concentrations of both orexin (95 pg/ml) and MCH peptide (6 pg/ml) in the CSF was reported (Wada et al., 2018).

We modified the sentence in the Discussion as follows:

â€œHowever, a patient with hypersomnia had low concentrations of both orexin (95 pg/ml) and MCH peptide (6 pg/ml) in the cerebrospinal fluid (Wada et al., 2018). Studying this case in detail will help to understand the role of MCH neurons on sleep/wakefulness in humans.

- Number of mice used in Figure 7, Figure 8 and Figure 9 is not shown.

We added the number of mice used in the legend of Figure 7, Figure 8 and Figure 9.

Figure 7: n=8, Figure 8: n=8, Figure 9: n=8.

- Figure 4. According to the figure legend, Figure 3A shows the "Relative EEG power in three strains of mice in the DOX(+) condition during the dark phase." Similarly, the main text (subsection 'Dual ablated mice frequently displayed short behavioral arrests with high spectral power in the δ and θ¸ bands of the EEG during wakefulness') also mentions three groups tested "in DOX(+) condition". But the legend in the Figure 4A (on the upper right) shows OXMC mice DOX(-), OXMC mice DOX(+), OX mice DOX(-). I suspect that the figure legend is correct?

We are sorry for the confusion. Figure 3A indicates representative traces of EEG and EMG with power spectral analysis of 3 different groups (OXMC DOX(-), OXMC DOX(+) and OX DOX(-)) after 4 weeks DOX (-) or DOX (+). Figure 4A showed FFT spectral analysis of 3 different groups (OXMC DOX(-), OXMC DOX(+) and OX DOX(-)) before DOX removal (0w).

To clarify, we have added the following sentence in the Figure 4 legend. "Relative EEG power in three strains of mice in the DOX(+) condition (before DOX removal) during the dark phase."

- Figure 4D. For better comparison, could the authors include a comparison of the EMG amplitude during the different states?

Thank you for your suggestion. We compared the EMG integral in each stage before and after DOX removal. The EMG integral did not significantly change after DOX removal. This data was added as Figure 4E. We added the following sentence in the Figure 4 legend. "(E) EMG integral of OXMC mice after DOX (-) condition."

- Figure 10. For better comparison, could the authors also compare the EMG amplitudes during DT sleep and NREM to REM transitions?

Thank you for your advice. According to reviewer's suggestion, we analyzed the EMG integral during DT sleep and NREM to REM transitions in OXMC DOX(-) 4w mice. We added the average EMG integral for 24 hours as an inset of Figure 10C. The EMG integral was not significantly different between DT sleep and the NREM to REM transitions.

- The finding that MCH neurons also suppress cataplexy is interesting and surprising. One question is whether this is specific to MCH neurons or general to other GABAergic neurons in the same region. I am not sure if it's feasible to add a control experiment by ablating a similar number of neurons in the lateral hypothalamus that are not necessarily MCH neurons. If not, the authors can perhaps add a discussion of this point.

This is an interesting question. Based on Mickelsen et al., (Mickelsen et al., 2017; Mickelsen et al., 2019), MCH neurons express Glutamate decarboxylase 1 (Gad1) but not vesicular GABA transporter (vGAT). In addition, MCH neurons express vesicular glutamate transporter 2 (vGlut2). This means that, although MCH neurons are able to produce GABA, they do not release GABA; rather, they release glutamate. We confirmed this using RNAscope in situ hybridization to label MCH, orexin, vGAT and vGlut2. Figure 1—figure supplement 1 indicate that most MCH neurons and orexin neurons co-localize vGlut2 (MCH neurons: 96.3 ± 3.1%, n=4; orexin neurons: 99.6 ± 0.2%, n=5) but not vGAT (MCH neurons: 5.1 ± 0.9%, n=4; orexin neurons: 1.7 ± 0.9%, n=3), suggesting that most of MCH and orexin neurons are glutamatergic neurons rather than GABAergic neurons. In the center of lateral hypothalamic area (indicated by dashed circle in the pictures), MCH neurons and orexin neurons are a minor subset of the entire glutamatergic neuron population (MCH neurons: 14.6 ± 2.4% (n=4); orexin neurons: 23.6 ± 3.2% (n=5)). These results suggest that ablation of a small number (10-15%) of additional MCH/glutamatergic neurons in the lateral hypothalamic area exacerbated narcolepsy phenotypes in orexin neuron-ablated mice. Although it would be very difficult to ablate a similar number of glutamatergic non-MCH neurons in the lateral hypothalamus, we believe that these facts emphasize the important role of MCH neurons to suppress cataplexy.

We added the following sentences in the Results section and Discussion section.

Results section

"In situ hybridization confirmed that MCH neurons and orexin neurons coexpressed vGlut2 mRNA but not vGAT mRNA. Figure 1—figure supplement 1 indicates that most MCH and orexin neurons are co-localized with vGlut2 (MCH neurons: 96.3 ± 3.1%, n=4; orexin neurons: 99.6 ± 0.2%, n=5) but not with vGAT (MCH neurons: 5.1 ± 0.9%, n=4; orexin neurons: 1.7 ± 0.9%, n=3), suggesting that most MCH and orexin neurons are glutamatergic rather than GABAergic neurons as also suggested by others (Mickelsen et al., 2017; Mickelsen et al., 2019). In the center of the LH (indicated by dotted circle), MCH neurons and orexin neurons were a relatively minor subset of the glutamatergic neuron population (MCH neurons: 14.6 ± 2.4% (n=4); orexin neurons: 23.6 ± 3.2% (n=5))."

Discussion section

"Our in situ hybridization results as well as previous reports (Mickelsen et al., 2017; Mickelsen et al., 2019) suggest that both orexin and MCH neurons are glutamatergic neurons. Among LH glutamatergic neurons, these two types of neurons are a relatively minor population (the proportion of MCH neurons and orexin neurons was 15% and 24%, respectively). These results suggest that ablation of a small number of additional LH glutamatergic neurons (10-15%) that co-express MCH exacerbates the narcolepsy phenotype in orexin neuronablated mice, underscoring the important role of MCH neurons to suppress cataplexy."